# Nonlinear Covariate Balance in Experimental Design

**Qing Chen** [1]  **Peng Zhang** [1]

## Abstract

We study experimental designs that balance nonlinear functions of covariates, extending classical methods that primarily target linear balance. Building on the Gram-Schmidt Walk (GSW) framework of Harshaw et al. (2024) for linear covariate balancing, we introduce a design that directly controls imbalance in nonlinear structure, including polynomial and more general smooth function classes. Like GSW, the proposed design retains sufficient robustness against model misspecification. Our implementation operates directly on a Gram matrix, avoiding the expensive step of explicitly constructing the nonlinear covariate expansions. We further accelerate the nonlinear design via a low-rank approximation of the Gram matrix, achieving runtimes comparable to the GSW of Harshaw et al. (2024) while preserving nonlinear covariate balance and robustness.

## 1. Introduction

Randomized Controlled Trials (RCTs) are the gold standard for estimating causal effects of new treatments and have been widely used in health care, economics, education, and digital platforms (Imbens & Rubin, 2015). In a two-arm RCT, units are randomly assigned to treatment or control. The *design* refers to the probability distribution of this random assignment. A simple design choice is an independent Bernoulli assignment, which is strongly robust because the assignment is fully random. However, pure randomization can cause substantial imbalance in pre-treatment *covariates* by chance, decreasing the precision of causal estimation when covariates are predictive of treatment outcomes. This motivates designs with restricted randomization that trade some robustness for improved covariate balance, including blocking (Fisher, 1935), matching (Greevy et al., 2004; Imai

et al., 2009; Bai et al., 2022), and rerandomization (Morgan & Rubin, 2012; Li et al., 2018; Li & Ding, 2020)

A recent important development by Harshaw et al. (2024) introduced the Gram-Schmidt Walk (GSW) design. It constructs a random assignment through a carefully guided random walk that keeps the induced covariate imbalance small at each step. GSW achieves a nearly *optimal* trade-off between covariate balance and robustness, yielding more accurate causal-effect estimates when outcomes depend linearly on covariates. This result has since motivated further work on algorithmic experimental design (Arbour et al., 2022; Chatterjee et al., 2025; Rao & Zhang, 2025; Chen et al., 2026).

**Nonlinear covariate structure matters.** A key limitation of GSW, noted by Harshaw et al. (2024), is that *it primarily targets balance for linear functions of covariates*. In many applications, covariate-outcome relationships are nonlinear: medical risk can vary nonlinearly with age, and effect modification may depend on covariate interactions (Morgan & Rubin, 2015; Bertsimas et al., 2015; Kallus, 2018). Even if two treatment arms are perfectly matched in covariate means or sums (and thus linear functions), they may still differ substantially in variances, interactions, and other higher-order structures (see an example in Figure 1), leading to poor estimation precision when the covariate-outcome relationship is nonlinear. Harshaw et al. (2024) remark that balancing nonlinear transforms requires manual feature choices, and point to a kernelized extension as a natural but out-of-scope direction. This raises a central question: *can we retain the appealing near-optimal robustness-balance trade-off of GSW while directly balancing rich nonlinear function classes?*

Prior work has argued for balancing richer covariate structures. For example, Bertsimas et al. (2015) advocate balancing both means and variances to improve similarity between treatment arms, and Kallus (2018) proposes balancing broad function classes using reproducing kernels (including linear, polynomial, exponential, and Gaussian kernels). A common approach in this line is to compute optimal balanced assignments by solving mixed-integer or semidefinite programs. While powerful in small experiments, such optimization-based designs can be computationally expensive as experiment size grows. More importantly, optimal designs may

---

[1]Department of Computer Science, Rutgers University, Piscataway, United States. Correspondence to: Peng Zhang <pz149@rutgers.edu>.

*Proceedings of the 43rd International Conference on Machine Learning*, Seoul, South Korea. PMLR 306, 2026. Copyright 2026 by the author(s).

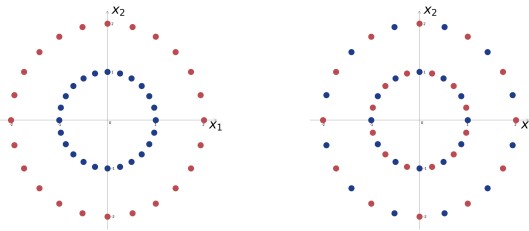

Figure 1. The points represent units with two covariates $x_1$ and $x_2$. In the left figure, the blue and red points have the same means in $x_1$ and $x_2$, but differ largely in their variances and covariances. The right figure shows a better blue-red assignment that balances both the means and the second-order statistics.

lack robustness when the assumed model class is misspecified (Krieger et al., 2019).

Unlike existing kernel-based optimal allocation methods, our goal is not to compute a deterministic or nearly deterministic optimal assignment by solving integer or semidefinite programs. Instead, we introduce nonlinear covariate balance into the randomized GSW framework, and retain the balance-robustness trade-off of Harshaw et al. (2024).

The key observation is that GSW can be implemented using only the unit-by-unit Gram matrix. We therefore construct a Gram matrix from a nonlinear feature map of the covariates and run Gram-GSW on this matrix. This balances higher-order covariate structure and smooth function classes without explicitly forming high- or infinite-dimensional feature expansions, making the method practical for moderate-to-large experiments.

We summarize our contributions as follows:

- **Method.** We show how to obtain *nonlinear* covariate balance by running Gram-GSW on a unit-by-unit Gram matrix induced by a nonlinear feature map. This construction balances higher-order covariate structure (e.g., quadratic/cubic interactions) *without* explicitly forming high- or infinite-dimensional feature expansions, and retains the robustness guarantee of GSW.

- **Theory.** We establish finite-sample guarantees for the nonlinear design, including mean-zero assignments, covariance bounds, and subgaussian discrepancy, in the induced feature space. Building on these guarantees, we derive covariate balance bounds for broad classes of smooth functions. We further establish finite-sample variance and tail bounds for the Horvitz-Thompson ATE estimator. We provide quantitative conditions under which including higher-order terms improves estimation precision.

- **Scalability.** To make nonlinear GSW practical at larger

scales, we develop a low-rank approximation implementation with explicit runtime-accuracy trade-offs, and characterize how approximation error propagates into covariance and MSE guarantees.

- **Empirical evaluation.** Through simulations spanning linear and nonlinear outcome models, we demonstrate that nonlinear GSW variants trade off some linear-outcome efficiency for improved nonlinear-outcome efficiency.

**Roadmap.** The remainder of the paper is organized as follows. Section 2 reviews the experimental setup and GSW. Section 3 gives the Gram-matrix implementation of GSW. Sections 4 and 5 introduce nonlinear GSW and prove its balance and estimation guarantees. Section 6 presents a low-rank implementation for scalability. Section 7 reports simulation results. We defer all proofs and additional results to the appendix.

## 2. Problem Setup

**Potential Outcomes, Covariates, and ATE.** We study RCTs with $n$ units, two treatment arms, and $d$-dimensional covariates. We follow the Neyman-Rubin potential outcomes framework (Rubin, 2005).

For each unit $i \in [n]$, let $a_i$ and $b_i$ denote the potential outcomes under treatment and control, respectively; only the outcome corresponding to the realized assignment is observed. Let $\mathbf{x}_i \in \mathbb{R}^d$ be the covariate vector of unit $i$, and let $\mathbf{X} = (\mathbf{x}_1, \ldots, \mathbf{x}_n)^\top \in \mathbb{R}^{n \times d}$ denote the covariate matrix. Throughout the paper, we treat the covariates and potential outcomes as fixed, and assume that all randomness arises from the treatment assignment. Let $\mathbf{z} = (z_1, \ldots, z_n)^\top \in \{-1, 1\}^n$ denote the random assignment vector, where $z_i = 1$ indicates treatment and $z_i = -1$ indicates control. We make the following standard assumptions.

**Assumption 2.1** (SUTVA (Imbens & Rubin, 2015)). Each unit's potential outcomes are unaffected by the treatment assignments of other units, and, for each unit and each treatment level, there are no alternative versions that could yield different potential outcomes.

**Assumption 2.2** (Assignment). The treatment assignment depends only on covariates, not on potential outcomes, i.e., $((a_1, b_1), \ldots, (a_n, b_n)) \perp\!\!\!\perp \mathbf{z} \mid (\boldsymbol{x}_1, \ldots, \boldsymbol{x}_n)$.

We want to estimate the *average treatment effect (ATE)* of the $n$ units, defined as

$$\tau \overset{\text{def}}{=} \frac{1}{n} \sum_{i=1}^n (a_i - b_i).$$

We use the Horvitz-Thompson (HT) estimator:

$$\hat{\tau} \stackrel{\text{def}}{=} \frac{1}{n}\left(\sum_{i:z_i=1}\frac{a_i}{\Pr(z_i=1)} - \sum_{i:z_i=-1}\frac{b_i}{\Pr(z_i=-1)}\right),$$

assuming $\Pr(z_i = 1) \in (0,1)$ for all $i$. In this paper, we will focus on the setting where $\Pr(z_i = 1) = \Pr(z_i = -1) = 1/2$ for each $i$. This can be satisfied, for example, by partitioning the $n$ units into two groups and then uniformly permuting the treatment arm labels across them.

**Lemma 2.3.** *The HT estimator is unbiased:* $\mathbb{E}[\hat{\tau}] = \tau$.

Thus, our goal is to construct a random $z$ to minimize the worst-case variance of $\hat{\tau}$, which equals mean-squared error (MSE) since $\hat{\tau}$ is unbiased.

**Estimation error.** Harshaw et al. (2024) show that the HT estimator has an error $\hat{\tau} - \tau = \frac{1}{n}\sum_{i=1}^{n}\mu_i z_i$, where $\mu_i = a_i + b_i$ for each $i$. Let $\boldsymbol{\mu} = (\mu_1, \ldots, \mu_n)^\top$ be the potential-outcome sum vector. The variance or the MSE is

$$\text{MSE}(\hat{\tau}) = \text{var}(\hat{\tau}) = \frac{1}{n^2}\,\boldsymbol{\mu}^\top\mathbb{E}[\boldsymbol{z}\boldsymbol{z}^\top]\boldsymbol{\mu}.$$

**Covariate balance and robustness.** Following Harshaw et al. (2024), we decompose the fixed potential-outcome sum vector $\boldsymbol{\mu}$ into its least-squares projection onto the column span of $\boldsymbol{X}$ and an orthogonal residual:

$$\boldsymbol{\mu} = \boldsymbol{X}\boldsymbol{\beta} + \boldsymbol{\epsilon}, \tag{1}$$

where

$$\boldsymbol{\beta} \in \arg\min_{\boldsymbol{\beta}' \in \mathbb{R}^d}\|\boldsymbol{\mu} - \boldsymbol{X}\boldsymbol{\beta}'\|, \qquad \boldsymbol{\epsilon} = \boldsymbol{\mu} - \boldsymbol{X}\boldsymbol{\beta}.$$

Both the vectors $\boldsymbol{\beta}$ and $\boldsymbol{\epsilon}$ are fixed but unknown. Under this decomposition, the MSE can be bounded as

$$\text{MSE}(\hat{\tau}) \leq \frac{2}{n^2}\big(\|\text{Cov}(\boldsymbol{X}^\top\boldsymbol{z})\|\,\|\boldsymbol{\beta}\|^2 + \|\text{Cov}(\boldsymbol{z})\|\,\|\boldsymbol{\epsilon}\|^2\big). \tag{2}$$

So, Harshaw et al. (2024) quantify covariate balance via $\|\text{Cov}(\boldsymbol{X}^\top\boldsymbol{z})\|$ and robustness via $\|\text{Cov}(\boldsymbol{z})\|$, and aim to find $z$ such that both operator norms are small. If the covariate-outcome relationship is nearly linear, the residual norm $\|\boldsymbol{\epsilon}\|$ is small, implying that improved covariate balance reduces the MSE.

The decomposition of the outcome sum vector $\boldsymbol{\mu}$ in Equation (1) can be extended by projecting $\boldsymbol{\mu}$ onto a richer linear space that includes nonlinear functions of the covariates, following Bertsimas et al. (2015) and Kallus (2018). For example, consider the class of polynomials of degree at most two,

$$\mathcal{F} = \text{Span}\{1, x_1, \ldots, x_d, x_1^2, \ldots, x_d^2, x_1 x_2, \ldots, x_{d-1}x_d\},$$

where $x_i$ denotes the $i$th coordinate of $\boldsymbol{x} \in \mathbb{R}^d$. We construct a matrix $\boldsymbol{\Psi} \in \mathbb{R}^{n \times \dim(\mathcal{F})}$ whose $i$th row consists of the evaluations of the basis functions in $\mathcal{F}$ at the covariate vector $\boldsymbol{x}_i$, and replace $\boldsymbol{X}$ with $\boldsymbol{\Psi}$ in Equations (1) and (2). When the covariate-outcome relationship is approximately quadratic, balancing $\boldsymbol{\Psi}$ yields a smaller MSE than balancing the original covariate matrix $\boldsymbol{X}$.

However, several important challenges remain. First, incorporating a large number of nonlinear covariate terms can dramatically increase the dimension of the features to be balanced, resulting in substantial computational cost; for example, including all quadratic terms increases the feature dimension from $d$ to $d^2$. Second, as observed by Harshaw et al. (2024), balancing many nonlinear features may worsen balance on linear ones, and it remains unclear whether the favorable statistical guarantees of GSW continue to hold and which nonlinear features should be balanced. This paper addresses both challenges.

## 3. Gram GSW and Feature GSW

We first review the GSW design (Harshaw et al., 2024). Then, we introduce two interchangeable implementations: one based on Gram matrices and one based on feature matrices. We use "feature" to refer to any transformation of covariates, and we use "covariate" to refer to the raw covariates.

The GSW design builds on the GSW algorithm of Bansal et al. (2019) in the context of algorithmic discrepancy theory. It takes as input a matrix $\boldsymbol{B} \in \mathbb{R}^{m \times n}$ or its Gram matrix $\boldsymbol{B}^\top\boldsymbol{B}$ and outputs a random vector $\boldsymbol{z} \in \{\pm 1\}^n$ such that the covariance matrix of $\boldsymbol{B}\boldsymbol{z}$ has small norm[1]. The $i$th column of $\boldsymbol{B}$ can be interpreted as an augmented or transformed covariate vector of the $i$th unit, and $\boldsymbol{z}$ represents an assignment of units into two treatment arms labeled 1 and $-1$. Thus, $\boldsymbol{B}\boldsymbol{z}$ is the difference between the sums of those covariate vectors across the two arms.

The central idea of GSW is to perform a random walk on the hypercube $[-1,1]^n$, starting from $\boldsymbol{z}_0 = \boldsymbol{0}$ and iteratively updating $\boldsymbol{z}_t = \boldsymbol{z}_{t-1} + \delta_t\boldsymbol{u}_t$, until $\boldsymbol{z}_t \in \{\pm 1\}^n$. Each update is designed to satisfy three properties: (i) $\boldsymbol{z}_t \in [-1,1]^n$; (ii) once a coordinate $\boldsymbol{z}_t(i)$ reaches $\pm 1$, it remains fixed thereafter; and (iii) the increment $\boldsymbol{B}(\boldsymbol{z}_t - \boldsymbol{z}_{t-1})$ stays close to $\boldsymbol{0}$, ensuring that the final covariance matrix of $\boldsymbol{B}\boldsymbol{z}$ has small norm. The key is how to choose the update direction $\boldsymbol{u}_t$. Algorithm 1 presents a generic GSW procedure, and Figure 2 shows an example with $n = 3$.

**DIRECTIONORACLE.** The subroutine in line 5 plays a central role. It computes the minimizer of the following

---

[1]GSW also guarantees that $\boldsymbol{B}\boldsymbol{z}$ is subgaussian.

**Algorithm 1** Generic GSW

**Require:** Feature matrix $\boldsymbol{B} \in \mathbb{R}^{m \times n}$ or Gram matrix $\boldsymbol{G} = \boldsymbol{B}^\top \boldsymbol{B} \in \mathbb{R}^{n \times n}$.

**Ensure:** A random assignment vector $\boldsymbol{z} \in \{\pm 1\}^n$.

1: Let $\boldsymbol{z}_0 = (0, 0, \ldots, 0)^\top$.
2: **for** $t = 1, 2, \ldots, n$ **do**
3:     Form the active set: $\mathcal{A}_t = \{i \in [n] : |\boldsymbol{z}_{t-1}(i)| < 1\}$.
4:     Select a pivot: $p_t = \max\{i \in \mathcal{A}_t\}$.
5:     Compute an update step direction:

$$\boldsymbol{u}_t(\mathcal{A}_t \setminus p_t) \leftarrow \text{DIRECTIONORACLE}(\boldsymbol{B} \text{ or } \boldsymbol{G}, \mathcal{A}_t, p_t),$$
$$\boldsymbol{u}_t(p_t) = 1, \quad \boldsymbol{u}_t(i) = 0, \text{ for all } i \notin \mathcal{A}_t$$

6:     Sample a random step size that is mean-zero and freezes at least one coordinate in $\mathcal{A}_t$:

$$\delta_t \leftarrow \text{STEPSIZEORACLE}(\boldsymbol{z}_{t-1}, \boldsymbol{u}_t).$$

7:     Update $\boldsymbol{z}_t = \boldsymbol{z}_{t-1} + \delta_t \boldsymbol{u}_t$.
8: **end for**
9: **Return** assignment vector $\boldsymbol{z}_n \in \{\pm 1\}^n$.

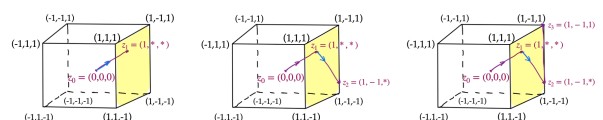

*Figure 2.* An example of GSW with $n = 3$: the random walk proceeds from $\boldsymbol{z}_0 = (0, 0, 0)$ to $\boldsymbol{z}_1 = (1, *, *)$, then to $\boldsymbol{z}_2 = (1, -1, *)$, and finally to $\boldsymbol{z}_3 = (1, -1, 1)$. At each step, one coordinate is frozen; the freezing order is determined by the random walk and can be arbitrary.

linear regression problem:

$$\boldsymbol{u}_t(\mathcal{A}_t \setminus p_t) = \arg\min_{\boldsymbol{u} \in \mathbb{R}^{\mathcal{A}_t \setminus p_t}} \left\| \boldsymbol{B}_{\mathcal{A}_t \setminus p_t} \boldsymbol{u} + \boldsymbol{b}_{p_t} \right\|^2, \quad (3)$$

where $\boldsymbol{B}_{\mathcal{A}_t \setminus p_t}$ denotes the submatrix of $\boldsymbol{B}$ formed by the columns indexed by $\mathcal{A}_t \setminus p_t$, and $\boldsymbol{b}_{p_t}$ is the $p_t$th column of $\boldsymbol{B}$. That is, the pivot column $\boldsymbol{b}_{p_t}$ is regressed onto the active columns $\boldsymbol{B}_{\mathcal{A}_t \setminus p_t}$. Intuitively, the algorithm seeks to freeze the pivot coordinate while leveraging the remaining active columns to offset the imbalance caused by freezing the pivot.

**STEPSIZEORACLE.** The subroutine in line 6 chooses a maximal feasible and mean-zero step along the update direction $\boldsymbol{u}_t$ that freezes at least one active coordinate. A standard implementation proceeds as follows. Define $\Delta = \{\delta \in \mathbb{R} : \boldsymbol{z}_{t-1} + \delta \boldsymbol{u}_t \in [-1, 1]^n\}$, and let $\delta^+ = \max \Delta$ and $\delta^- = -\min \Delta$. The step size $\delta_t \in \{\delta^+, -\delta^-\}$ is then sampled at random, with probabilities proportional to $\delta^-$ and $\delta^+$, respectively.

**Augmented covariate matrix $B$ in the GSW design for balance-robustness tradeoff.** Given a balance-robustness

tradeoff parameter $\phi \in (0, 1)$, define the augmented matrix

$$\boldsymbol{B} \stackrel{\text{def}}{=} \begin{pmatrix} \sqrt{\phi}\, \boldsymbol{I}_n \\ \sqrt{1 - \phi}\, \xi^{-1}\, \boldsymbol{X}^\top \end{pmatrix}, \qquad \xi \stackrel{\text{def}}{=} \max_{i \in [n]} \|\boldsymbol{x}_i\|_2. \quad (4)$$

Larger values of $\phi$ emphasize robustness, and smaller values emphasize covariate balance. As discussed in Section 2 and noted by Harshaw et al. (2024), the covariate matrix $\mathbf{X}$ can be replaced by a richer matrix $\boldsymbol{\Psi}$ that includes nonlinear transformations of the covariates. We will elaborate on this extension in the following sections.

### 3.1. Two Interchangeable Direction Oracles

We describe two implementations of DIRECTIONORACLE, the key subroutine in line 5 of Algorithm 1. The regression problem in Equation (3) has a closed-form solution via the normal equations. Using the matrix $\boldsymbol{B}$ defined in Equation (4), the vector $\boldsymbol{u}_t$ can be written as

$$\boldsymbol{u}_t(\mathcal{A}_t \setminus p_t) = -(\lambda \boldsymbol{I} + \boldsymbol{X}_t \boldsymbol{X}_t^\top)^{-1} \boldsymbol{X}_t \boldsymbol{x}_{p_t}, \quad (5)$$

where $\lambda = \frac{\xi^2 \phi}{1 - \phi}$ and $\boldsymbol{X}_t$ is the submatrix of $\boldsymbol{X}$ restricted to the rows indexed by $\mathcal{A}_t \setminus p_t$.

**Gram GSW.** Equation (5) depends only on the pairwise inner products of covariate vectors $\{\langle \boldsymbol{x}_i, \boldsymbol{x}_j \rangle : i, j \in [n]\}$. As a result, GSW can be implemented using the Gram matrix $\boldsymbol{K} = \boldsymbol{X} \boldsymbol{X}^\top$ via

$$\boldsymbol{u}_t(\mathcal{A}_t \setminus p_t) = -(\lambda \boldsymbol{I} + \boldsymbol{K}_t)^{-1} \boldsymbol{k}_{p_t},$$

where $\boldsymbol{K}_t$ is the submatrix of $\boldsymbol{K}$ restricted to the rows and columns indexed by $\mathcal{A}_t \setminus p_t$ and $\boldsymbol{k}_{p_t}$ is the $p_t$-th column of $\boldsymbol{K}$ restricted to the same index set. We refer to Algorithm 1, run with input $\phi$ and the Gram matrix $\boldsymbol{K}$, which together build the implicit Gram matrix of $\boldsymbol{B}$, and with DIRECTIONORACLE implemented via the above equation, as *Gram GSW$(\phi, \boldsymbol{K})$*.

Gram GSW is particularly useful when the feature dimension is large, for example, when many nonlinear transformations are included, provided that the corresponding inner products can be constructed efficiently. It avoids explicitly building the high-dimensional feature matrix.

**Feature GSW.** When $d \ll n$, Harshaw et al. (2024) shows that $\boldsymbol{u}_t$ can be computed more efficiently via

$$\boldsymbol{u}_t(\mathcal{A}_t \setminus p_t) = -\frac{1}{\lambda} \boldsymbol{X}_t \left( \boldsymbol{I} - (\boldsymbol{X}_t^\top \boldsymbol{X}_t + \lambda \boldsymbol{I})^{-1} \boldsymbol{X}_t^\top \boldsymbol{X}_t \right) \boldsymbol{x}_{p_t},$$

This formulation operates directly on the covariate (feature) matrix $\boldsymbol{X}$. We refer to Algorithm 1, run with input $\phi$ and the covariate matrix $\boldsymbol{X}$, which together build the implicit augmented matrix $\boldsymbol{B}$, and with DIRECTIONORACLE implemented via the above equation, as *Feature GSW$(\phi, \boldsymbol{X})$*.

**Lemma 3.1.** *Assume that $\phi \in (0,1)$. Under the same randomness on sampling step sizes, Feature GSW run on $\boldsymbol{B}$ and Gram GSW run on $\boldsymbol{G} = \boldsymbol{B}^\top \boldsymbol{B}$ produce the same distribution of the assignment vector $\boldsymbol{z}$.*

Gram GSW admits a cubic-time implementation by computing a Cholesky factorization of $\boldsymbol{K}$ and maintaining it through rank-one updates, following a similar approach in Harshaw et al. (2024).

**Lemma 3.2.** *Gram GSW on a Gram matrix $\boldsymbol{K} \in \mathbb{R}^{n \times n}$ can be implemented in time $O(n^3)$ and in space $O(n^2)$.*

## 4. Nonlinear GSW

We introduce Nonlinear GSW. The key idea is to apply Gram GSW after mapping each raw covariate vector to a nonlinear feature space. This feature map includes tensor powers of the covariates, so balancing in the feature space controls higher-order covariate structure such as quadratic and cubic moments. Since Gram GSW only requires pairwise inner products, the resulting design can be implemented through the induced Gram matrix without explicitly forming the high-dimensional feature vectors.

### 4.1. Feature Hilbert Space

For each $k \geq 1$, let $\mathcal{H}_k \stackrel{\text{def}}{=} (\mathbb{R}^d)^{\otimes k}$, equipped with the standard Frobenius inner product on tensors:

$$\langle \boldsymbol{u}, \boldsymbol{v} \rangle_{\mathcal{H}_k} = \sum_{j_1, \ldots, j_k} \boldsymbol{u}(j_1, \ldots, j_k)\, \boldsymbol{v}(j_1, \ldots, j_k)$$

and $\|\boldsymbol{u}\|_{\mathcal{H}_k}^2 = \langle \boldsymbol{u}, \boldsymbol{u} \rangle_{\mathcal{H}_k}$. Finally, set $\mathcal{H}_0 = \mathbb{R}$ with $\langle u, v \rangle_{\mathcal{H}_0} = uv$. Define the Hilbert direct sum

$$\mathcal{H} \stackrel{\text{def}}{=} \bigoplus_{k \geq 0} \mathcal{H}_k = \left\{ (\boldsymbol{u}_k)_{k \geq 0} : \boldsymbol{u}_k \in \mathcal{H}_k, \sum_{k \geq 0} \|\boldsymbol{u}_k\|_{\mathcal{H}_k}^2 < \infty \right\}$$

with inner product $\langle \boldsymbol{u}, \boldsymbol{v} \rangle_{\mathcal{H}} \stackrel{\text{def}}{=} \sum_{k \geq 0} \langle \boldsymbol{u}_k, \boldsymbol{v}_k \rangle_{\mathcal{H}_k}$.

**Feature map.** Let $\xi \stackrel{\text{def}}{=} \max_{i \in [n]} \|\boldsymbol{x}_i\|_2 > 0$, and let $\boldsymbol{\alpha} = \{\alpha_k\}_{k \geq 0}$ with $\alpha_k \in [0, 1]$ satisfying $\sum_{k \geq 0} \alpha_k \leq 1$. Define the feature map $\psi : \mathbb{R}^d \to \mathcal{H}$ by

$$\psi_{\boldsymbol{\alpha}}(\boldsymbol{x}) \stackrel{\text{def}}{=} \left( \sqrt{\alpha_0},\ \frac{\sqrt{\alpha_1}}{\xi}\, \boldsymbol{x},\ \frac{\sqrt{\alpha_2}}{\xi^2}\, \boldsymbol{x}^{\otimes 2},\ \ldots,\ \frac{\sqrt{\alpha_k}}{\xi^k}\, \boldsymbol{x}^{\otimes k},\ \ldots \right),$$

which consists of tensor products of all orders formed from the coordinates of $\boldsymbol{x}$. The parameter $\alpha_0 > 0$ includes the constant feature balance, so the design penalizes imbalance in $\sum_{i=1}^n z_i$, the treated-control group-size difference. GSW does not force this difference to be zero. When the context is clear, we omit the subscript $\boldsymbol{\alpha}$ for simplicity.

Observe that $\|\boldsymbol{x}^{\otimes k}\|_{\mathcal{H}_k} = \|\boldsymbol{x}\|_2^k$ for all $k \geq 0$, with the convention $\boldsymbol{x}^{\otimes 0} = 1$. Thus, if $\|\boldsymbol{x}\|_2$ is bounded, then $\|\psi_{\boldsymbol{\alpha}}(\boldsymbol{x})\|_{\mathcal{H}}$ is also bounded.

*Claim* 4.1. Suppose $\|\boldsymbol{x}\|_2 \leq \xi$ and $\sum_{k \geq 0} \alpha_k \leq 1$. Then, $\|\psi_{\boldsymbol{\alpha}}(\boldsymbol{x})\|_{\mathcal{H}}^2 \leq 1$ and thus $\psi_{\boldsymbol{\alpha}}(\boldsymbol{x}) \in \mathcal{H}$.

Next, define the linear operator $\boldsymbol{\Psi} : \mathcal{H} \to \mathbb{R}^n$ by

$$(\boldsymbol{\Psi}\boldsymbol{\beta})_i \stackrel{\text{def}}{=} \langle \psi_{\boldsymbol{\alpha}}(\boldsymbol{x}_i), \boldsymbol{\beta} \rangle_{\mathcal{H}}, \qquad i \in [n],$$

and let $\boldsymbol{\Psi}^* : \mathbb{R}^n \to \mathcal{H}$ denote its adjoint, given by $\boldsymbol{\Psi}^* \boldsymbol{z} = \sum_{i=1}^n z_i \psi_{\boldsymbol{\alpha}}(\boldsymbol{x}_i)$. We further define the augmented linear map $\boldsymbol{B} : \mathbb{R}^n \to \mathbb{R}^n \oplus \mathcal{H}$ as $\boldsymbol{B}\boldsymbol{z} \stackrel{\text{def}}{=} \left( \sqrt{\phi}\, \boldsymbol{z},\ \sqrt{1-\phi}\, \boldsymbol{\Psi}^* \boldsymbol{z} \right)$. The associated Gram matrix is $\boldsymbol{K} = \boldsymbol{\Psi}\boldsymbol{\Psi}^* \in \mathbb{R}^{n \times n}$, with entries

$$\boldsymbol{K}(i,j) = \langle \psi_{\boldsymbol{\alpha}}(\boldsymbol{x}_i), \psi_{\boldsymbol{\alpha}}(\boldsymbol{x}_j) \rangle_{\mathcal{H}} = \sum_{k \geq 0} \alpha_k \frac{\langle \boldsymbol{x}_i, \boldsymbol{x}_j \rangle^k}{\xi^{2k}} < \infty.$$

Importantly, $\boldsymbol{K}$ depends only on the inner products of the raw covariate vectors. With suitable choices of the coefficients $\{\alpha_k\}$, the above series can be evaluated in closed form. For example, taking only finitely many $\alpha_k$ to be nonzero yields a polynomial kernel, and setting $\alpha_k = e^{-1}/k!$ gives $\boldsymbol{K}(i,j) = \exp(\xi^{-2} \langle \boldsymbol{x}_i, \boldsymbol{x}_j \rangle - 1)$, a normalized exponential kernel. Consequently, $\boldsymbol{K}$ can be computed efficiently without explicitly constructing the high- or infinite-dimensional feature vectors $\psi(\boldsymbol{x}_i)$. We run Gram GSW on $\phi$ and $\boldsymbol{K}$.

GSW in Harshaw et al. (2024) corresponds to the special linear case where $\alpha_1 = 1$ and $\alpha_k = 0$ for all $k \neq 1$. The following lemma generalizes their results to nonlinear cases.

**Lemma 4.2.** *Let $\boldsymbol{z}$ be the assignment returned by Gram GSW$(\phi, \boldsymbol{K})$. Then, $\mathbb{E}[\boldsymbol{z}] = \boldsymbol{0}$, and $\mathrm{Cov}(\boldsymbol{z}) \preccurlyeq (\phi\boldsymbol{I} + (1-\phi)\boldsymbol{K})^{-1}$. Moreover, $\boldsymbol{B}\boldsymbol{z}$ is subgaussian[2] in $\mathbb{R}^n \oplus \mathcal{H}$ with variance proxy $1$.*

Lemma 4.2 is tight. If $\alpha_0 = 0$, $\sum_{k \geq 1} \alpha_k = 1$ and the raw covariate vectors $\{\boldsymbol{x}_i\}$ are mutually orthogonal and have unit norm, then the feature vectors $\{\psi(\boldsymbol{x}_i)\}$ are also mutually orthogonal and thus $\boldsymbol{K} = \boldsymbol{I}$. In this case, the entries of $\boldsymbol{z}$ are independent and uniform, and $\mathrm{Cov}(\boldsymbol{z}) = \boldsymbol{I}$.

### 4.2. Balancing Smooth Functions of Covariates

We can balance a broad class of smooth functions of covariates, using the feature map $\psi$, the linear map $\boldsymbol{\Psi}$, and the Gram matrix $\boldsymbol{K}$, defined in Section 4.1.

---

[2]We say that a mean-zero random $\boldsymbol{y}$ in a Hilbert space $\mathcal{K}$ is subgaussian with variance proxy $\sigma^2$ (or $\sigma$-subgaussian) if, for every $\boldsymbol{\theta} \in \mathcal{K}$ and every $\lambda \in \mathbb{R}$, $\mathbb{E}\left[ \exp\left(\lambda \langle \boldsymbol{\theta}, \boldsymbol{y} \rangle_{\mathcal{K}}\right) \right] \leq \exp\left( \frac{\lambda^2 \sigma^2 \|\boldsymbol{\theta}\|_{\mathcal{K}}^2}{2} \right)$. Equivalently, every one-dimensional projection $\langle \boldsymbol{\theta}, \boldsymbol{y} \rangle_{\mathcal{K}}$ is subgaussian with variance proxy at most $\sigma^2 \|\boldsymbol{\theta}\|_{\mathcal{K}}^2$.

Consider $f : \mathbb{R}^d \to \mathbb{R}$ in $C^{m+1}$ being a finite-smoothness function for a fixed integer $m \geq 1$. We can split $f$ into a "degree $\leq m$" Taylor polynomial plus a remainder term. For each $k \leq m+1$, let $D^k f(\mathbf{0}) \in \mathcal{H}_k$ denote the $k$th derivative tensor at $\mathbf{0}$ (as a symmetric $k$-linear form). Then,

$$f(\boldsymbol{x}) = \sum_{k=0}^{m} \frac{1}{k!} D^k f(\mathbf{0})[\boldsymbol{x}^{\otimes k}] + R_m(\boldsymbol{x}),$$

where $R_m(\boldsymbol{x})$ is the remainder satisfying the standard bound

$$|R_m(\boldsymbol{x})| \leq \frac{1}{(m+1)!} \sup_{t \in [0,1]} \left\| D^{m+1} f(t\boldsymbol{x}) \right\|_{\mathrm{op}} \|\boldsymbol{x}\|_2^{m+1}.$$

**Lemma 4.3.** *Let $\boldsymbol{z}$ be the assignment returned by Gram $GSW(\phi, \boldsymbol{K})$ where $\alpha_k > 0$ for $0 \leq k \leq m$. Let $f \in C^{m+1}$ for some $m \in \mathbb{Z}_{\geq 1}$. Let $S_f \overset{\text{def}}{=} \sum_{i=1}^{n} z_i f(\boldsymbol{x}_i)$ be the imbalance of $f$ on covariates. Let*

$$T_1 \overset{\text{def}}{=} \sum_{k=0}^{m} \frac{\xi^{2k}}{(k!)^2 \alpha_k} \left\| D^k f(\mathbf{0}) \right\|_{\mathcal{H}_k}^2, \quad T_2 \overset{\text{def}}{=} \sum_{i=1}^{n} R_m(\boldsymbol{x}_i)^2.$$

*Then, $\mathbb{E}[S_f] = 0$, $\mathrm{var}(S_f) \leq \frac{T_1}{1-\phi} + \frac{T_2}{\phi}$, and $S_f$ is subgaussian with variance proxy $\frac{T_1}{1-\phi} + \frac{T_2}{\phi}$.*

Intuitively, the degree $\leq m$ Taylor polynomial part of $f$ is balanced by $\boldsymbol{\Psi}$ (covariate balance), and the remainder part is balanced by the randomness of $\boldsymbol{z}$. So, the term $\frac{T_1}{1-\phi}$ is what we gain by balancing the first $m$ tensor moments (controlled by $\psi_{\boldsymbol{\alpha}}$); the term $\frac{T_2}{\phi}$ is what we pay for whatever the degree-$m$ Taylor polynomial misses (handled by randomness).

Below, we provide concrete examples of smooth $f$.

**Polynomials.** $f$ is a degree-$m$ polynomial. Then, $R_m \equiv 0$, and thus $T_2 = 0$, and the whole imbalance is controlled purely through the balanced tensor moments. Consider $f(\boldsymbol{x}_i) = \sum_{k=0}^{m} \langle \boldsymbol{A}_k, \boldsymbol{x}_i^{\otimes k} \rangle_{\mathcal{H}_k}$. By Lemma 4.3, $S_f$ is $(\frac{T_1}{1-\phi})^{1/2}$-subgaussian, where $T_1 = \sum_{k \geq 0} \frac{\xi^{2k}}{\alpha_k} \|\boldsymbol{A}_k\|_{\mathcal{H}_k}^2$ is the squared norm of the polynomial coefficients weighted by $\boldsymbol{\alpha}$ and $\xi$.

**Multivariate smooth functions.** Lemma 4.3 applies directly to any $C^{m+1}$ function whose $(m+1)$-st derivatives are bounded on a ball containing the observed covariates. In this case, the degree-$m$ Taylor part is controlled by the feature imbalance through $T_1$, while the Taylor remainder is controlled through $T_2$. Thus, for any fixed $m$, $S_f$ is subgaussian whenever $T_1$ and $T_2$ are finite.

This covers many standard nonlinear response models on bounded covariate sets, including additive models, finite-order interaction models, smooth links of linear or quadratic indices, and analytic single-index models $f(\boldsymbol{x}) = g(\boldsymbol{a}^\top \boldsymbol{x})$ with smooth $g$, such as $g = \exp, \sin, \cos, \sinh, \cosh$. For

radial examples, $f(\boldsymbol{x}) = g(\|\boldsymbol{x} - \boldsymbol{a}\|_2^2)$ is smooth under mild smoothness conditions on $g$.

**Choosing $\boldsymbol{\alpha}$.** We consider that feature weights $\boldsymbol{\alpha} = \{\alpha_k\}_{k \geq 0}$ are chosen by experimenters based on domain knowledge. The feature weight $\alpha_k$ determines how much priority is placed on balancing the $k$th covariate moment. We recommend placing a large weight on the first moment (i.e., choosing $\alpha_1$ to be relatively large), since balancing on raw covariates is important across a wide range of settings. If low-order nonlinearities or interactions are expected, we recommend a low-degree polynomial design such as degree 2 or 3. Even a small positive $\alpha_2$ or $\alpha_3$ can greatly reduce second- or third-moment imbalance, with only a minor increase in first-moment imbalance. We use an exponential kernel when one wants to include all moments, and it is considerably more expressive.

# 5. Finite-Sample Properties

**Proposition 5.1.** *Let $\hat{\tau}$ be the HT estimator under nonlinear GSW. Then, $\mathbb{E}[\hat{\tau}] = \tau$, and*

$$n\,\mathrm{var}(\hat{\tau}) \leq L_{\phi,\alpha} \overset{\text{def}}{=}$$
$$\min_{\boldsymbol{\beta} \in \mathcal{H}} \left( \frac{1}{\phi n} \|\boldsymbol{\mu} - \boldsymbol{\Psi}\boldsymbol{\beta}\|_2^2 + \frac{1}{(1-\phi)n} \|\boldsymbol{\beta}\|_{\mathcal{H}}^2 \right). \quad (6)$$

*Moreover, $\Pr\left( |\hat{\tau} - \tau| \geq \gamma \right) \leq 2 \exp\left( -\frac{\gamma^2 n}{2 L_{\phi,\alpha}} \right)$ for any $\gamma \geq 0$.*

Proposition 5.1 extends the ridge-regression upper bound on the variance of the HT estimator from linear GSW to nonlinear GSW. We refer to $L_{\phi,\alpha}$ as the corresponding ridge upper bound.

Including additional higher-order covariate moments in $\boldsymbol{\Psi}$ can reduce the approximation error term $\|\boldsymbol{\mu} - \boldsymbol{\Psi}\boldsymbol{\beta}\|_2^2$ by enlarging the covariate space that is used to explain the outcomes. However, the coefficients of $\boldsymbol{\beta}$ corresponding to $\boldsymbol{x}^{\otimes k}$ scale inversely with their associated weights $\alpha_k$, so incorporating many higher-order moments (and thus small $\alpha_k$ values) can increase the regularization term $\|\boldsymbol{\beta}\|_{\mathcal{H}}^2$. This trade-off raises the question of when, and to what extent, higher-order covariate terms should be included.

The following corollary shows that the ridge upper bound for nonlinear GSW is at most a factor of $1/\alpha_1$ larger than the corresponding ridge upper bound for linear GSW.

**Corollary 5.2.** *Let $L_H$ denote the ridge upper bound in Harshaw et al. (2024), so that $n\mathrm{var}(\hat{\tau}) \leq L_H$ under their GSW design. If $\alpha_1 > 0$, then $L_{\phi,\alpha} \leq L_H/\alpha_1$.*

## 5.1. When Balancing Second-Order Terms Improves Precision

We establish conditions under which balancing second-order covariate moments improves the precision of the HT estimator. The same result naturally extends to higher moments.

We assume only $\alpha_1$ and $\alpha_2$ are nonzero and set $\alpha_1 = \alpha$ and $\alpha_2 = 1 - \alpha$ for $\alpha \in (0, 1)$. The resulting feature map is $\boldsymbol{\Psi} = \left( \sqrt{\alpha} \xi^{-1} \boldsymbol{X} \quad \sqrt{1-\alpha} \xi^{-2} \boldsymbol{X}^{\otimes 2} \right)$, where $\boldsymbol{X}^{\otimes 2} \in \mathbb{R}^{n \times d^2}$ has rows given by entries of $\boldsymbol{x}_i^{\otimes 2}$. Our goal is to characterize the value $\alpha^*$ that minimizes the ridge upper bound $L_{\phi,\alpha}$ for any fixed $\phi \in (0, 1)$.

**Lemma 5.3.** *Fix $\phi \in (0, 1)$ and define*

$$\boldsymbol{K}_1 \stackrel{\text{def}}{=} \boldsymbol{X}\boldsymbol{X}^\top, \qquad \boldsymbol{K}_2 \stackrel{\text{def}}{=} \boldsymbol{X}^{\otimes 2}(\boldsymbol{X}^{\otimes 2})^\top,$$

$$\boldsymbol{w}(\alpha) \stackrel{\text{def}}{=} \left(\lambda \boldsymbol{I} + \alpha \boldsymbol{K}_1 + (1-\alpha)\xi^{-2} \boldsymbol{K}_2\right)^{-1} \boldsymbol{\mu}, \; \lambda \stackrel{\text{def}}{=} \frac{\phi \xi^2}{1 - \phi}.$$

*Let $\alpha^*$ denote a minimizer of $L_{\phi,\alpha}$ over $[0, 1]$. Let $g(\alpha) \stackrel{\text{def}}{=} \boldsymbol{w}(\alpha)^\top (\boldsymbol{K}_1 - \xi^{-2}\boldsymbol{K}_2)\boldsymbol{w}(\alpha)$. If $g(1) < 0$, then $\alpha^* < 1$; if $g(0) > 0$, then $\alpha^* > 0$. Moreover, if both conditions hold, then $\alpha^*$ is the unique solution to $g(\alpha) = 0$.*

Intuitively, $\boldsymbol{w}(1)$ is the dual vector that identifies the directions in sample space most relevant for approximating $\boldsymbol{\mu}$ under the ridge model with $\alpha = 1$. Decreasing $\alpha$ shifts weight from the linear kernel $\boldsymbol{K}_1$ to the quadratic kernel $\boldsymbol{K}_2$. This shift decreases the ridge upper bound $L_{\phi,\alpha}$ precisely when, along the direction $\boldsymbol{w}(1)$, the quadratic kernel contributes more than the linear kernel after the $\xi^{-2}$ scaling.

# 6. Faster Implementation: Low-Rank GSW

We further accelerate nonlinear GSW via a low-rank approximation of the Gram matrix, achieving runtimes comparable to the linear GSW of Harshaw et al. (2024) while preserving higher-order covariate balance.

Specifically, we approximate the Gram matrix $\boldsymbol{K} = \boldsymbol{\Psi}\boldsymbol{\Psi}^\top \in \mathbb{R}^{n \times n}$ with a rank-$k$ matrix positive semidefinite $\tilde{\boldsymbol{K}}$, where $k \ll n$. Let $\tilde{\boldsymbol{\Psi}} \in \mathbb{R}^{n \times k}$ denote a factor such that $\tilde{\boldsymbol{\Psi}}\tilde{\boldsymbol{\Psi}}^\top = \tilde{\boldsymbol{K}}$. We then run Feature GSW with $\phi$ and $\tilde{\boldsymbol{\Psi}}$, reducing the runtime to $O(n^2 k)$, which is substantially smaller than $O(n^3)$. Let $\tilde{\boldsymbol{z}}$ denote the resulting treatment assignment. We present a pseudo-code in Algorithm 2.

**Lemma 6.1.** *Let $\tilde{\boldsymbol{K}}$ be a symmetric PSD rank-$k$ approximation to $\boldsymbol{K}$ satisfying $\|\tilde{\boldsymbol{K}} - \boldsymbol{K}\|_2 \le \epsilon_k < \frac{\phi}{1-\phi}$. Let $\tilde{\boldsymbol{z}}$ be the assignment returned by Low-Rank GSW using $\tilde{\boldsymbol{K}}$. Then, $\mathbb{E}[\tilde{\boldsymbol{z}}] = 0$, and*

$$\text{Cov}(\tilde{\boldsymbol{z}}) \preccurlyeq \left((\phi - (1-\phi)\epsilon_k)\boldsymbol{I} + (1-\phi)\boldsymbol{K}\right)^{-1}.$$

---

**Algorithm 2** Low-Rank GSW Design

**Require:** A Gram matrix $\boldsymbol{K} \in \mathbb{R}^{n \times n}$, a tradeoff parameter $\phi \in (0, 1)$, and a low-rank parameter $k \in \mathbb{Z}_+$.
**Ensure:** A random assignment vector $\tilde{\boldsymbol{z}} \in \{\pm 1\}^n$.
1: Compute a rank-$k$ symmetric PSD matrix $\tilde{\boldsymbol{K}} \in \mathbb{R}^{n \times n}$ that approximates $\boldsymbol{K}$, and a factorization

$$\tilde{\boldsymbol{K}} = \tilde{\boldsymbol{\Psi}}\tilde{\boldsymbol{\Psi}}^\top, \quad \text{where } \tilde{\boldsymbol{\Psi}} \in \mathbb{R}^{n \times k}.$$

2: **Return** $\tilde{\boldsymbol{z}} \leftarrow$ Feature GSW on $(\phi, \tilde{\boldsymbol{\Psi}})$ without additional normalization on $\tilde{\boldsymbol{\Psi}}$.

---

**Lemma 6.2.** *Let $\hat{\tau}_{\text{LR}}$ be the HT estimator under low-rank nonlinear GSW with $\tilde{\boldsymbol{K}}$ satisfying the conditions in Lemma 6.1. Then, $\mathbb{E}[\hat{\tau}_{\text{LR}}] = \tau$, and*

$$n\,var(\hat{\tau}_{\text{LR}}) \le \min_{\boldsymbol{\beta} \in \mathcal{H}} \left( \frac{1}{\phi'n} \|\boldsymbol{\mu} - \boldsymbol{\Psi}\boldsymbol{\beta}\|_2^2 + \frac{1}{(1-\phi)n} \|\boldsymbol{\beta}\|_{\mathcal{H}}^2 \right),$$

*where $\phi' = \phi - (1 - \phi)\epsilon_k$.*

A widely used approach for constructing low-rank approximations of Gram matrices is the Nyström method (Williams & Seeger, 2000; Drineas & Mahoney, 2005; Gittens & Mahoney, 2016; Musco & Musco, 2017). It uses only $O(nk)$ kernel evaluations and incurs an additional $O(nk^2)$ cost to form the factorization. Consequently, the overall runtime of Algorithm 2 is $O(n^2 k)$.

The low-rank parameter $k$ controls a speed-accuracy tradeoff: larger $k$ gives a more accurate approximation of the nonlinear Gram matrix $\boldsymbol{K}$, but also increases computational cost. A useful guide for choosing $k$ is the spectrum of $\boldsymbol{K}$. For the normalized moment Gram matrices $\hat{\boldsymbol{K}}_\ell = \xi^{-2\ell}\boldsymbol{X}^{\otimes \ell}(\boldsymbol{X}^{\otimes \ell})^\top$, we have $\text{tr}(\hat{\boldsymbol{K}}_{\ell+1}) \le \text{tr}(\hat{\boldsymbol{K}}_\ell)$ for all $\ell \ge 1$. Thus, higher-order moment matrices do not carry more total spectral mass than lower-order ones; instead, they spread no more mass over a larger feature space. If the higher-order spectrum is concentrated, a moderate rank can capture its leading directions. If the spectrum is diffuse, the weighted eigenvalues of $\alpha_\ell \hat{\boldsymbol{K}}_\ell$ may fall below the error tolerance in Lemma 6.1 and can be omitted with little effect on the covariance bound. The difficult case is a long plateau of eigenvalues above this tolerance, in which case a larger $k$ might be needed. In our experiments, we use $k = O(d)$ as a computationally efficient default, and Appendix B.5 reports a sensitivity study over several choices of $k$.

# 7. Experiments

We empirically investigate Nonlinear GSW under different choices of the coefficients $\{\alpha_k\}$. Specifically, we consider: (i) GSW(1), the linear GSW with $\alpha_1 = 1$, corresponding to the original GSW of Harshaw et al. (2024); (ii) GSW(2),

a quadratic GSW with $\alpha_1 = 2/3$ and $\alpha_2 = 1/3$; (iii) GSW(3), a cubic GSW with $\alpha_1 = 1/2$, $\alpha_2 = 1/4$, and $\alpha_3 = 1/4$; and (iv) GSW($\infty$), an exponential GSW with $\alpha_k = e^{-1}/k!$ for all $k \geq 0$. For each GSW variant, we evaluate multiple values of the balance–robustness parameter $\phi \in \{0.01, 0.1, 0.5, 0.9, 0.99\}$. Smaller $\phi$ emphasizes covariate balance, and larger $\phi$ emphasizes robustness.

We implement GSW on the full Gram matrix for small $n \leq 100$ and on a rank-$2d$ approximation for $n > 100$.

We compare these designs with commonly used benchmark designs, including Bernoulli, Complete Randomization (CR), Blocking, Pairwise Matching, and Rerandomization (Rerand)[3]. We do not include the optimal kernel allocation designs of Kallus (2018), as their runtime becomes prohibitive even for moderately sized experiments (e.g., $n \geq 40$), see Figure 3, and these designs are not robust to model misspecification.

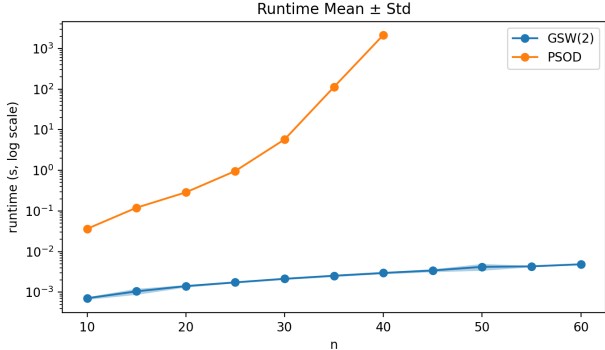

**Figure 3.** Runtime comparison between GSW(2) and PSOD (Kallus, 2018) for $d = 5$.

Our codes are available at https://github.com/pengzhang91/nonlinear.

**Data generating process.** Covariate vectors $\boldsymbol{x}_1, \ldots, \boldsymbol{x}_n \in \mathbb{R}^d$ are independently drawn from the standard normal distribution $\mathcal{N}(0, \boldsymbol{I}_d)$. The potential outcome vector $\boldsymbol{\mu} = (\mu_1, \ldots, \mu_n)$ where each $\mu_i = a_i + b_i$ is generated according to $\mu_i = f(\boldsymbol{x}_i) + \epsilon_i$, where $f$ is a fixed but unknown function and $\epsilon_i \sim \mathcal{N}(0, 0.1^2)$. We consider linear, quadratic, cubic, and sinusoidal choices of $f$, following Kallus (2018). Because the MSE depends on potential outcomes only through $\boldsymbol{\mu} = \boldsymbol{a} + \boldsymbol{b}$, we generate $\boldsymbol{\mu}$ directly. More details are provided in Appendix B.

We report simulation results for $d = 10$ and sample sizes $n \in \{30, 100, 200, 500\}$. Additional experiments with a larger covariate dimension $d = 30$ are reported in Appendix B. For each $n$, we evaluate multiple designs using

[3]Using the Mahalanobis distance with a $1\%$ exact acceptance probability.

100 independently generated covariate realizations. For each GSW variant and choice of the parameter $\phi$, we run 1,000 simulations and compute empirical robustness, covariate moment imbalance, and empirical MSE across four outcome-generating models.

**Robustness and covariate balance.** Table 2 shows the robustness metric and different covariate metrics for $n = 30, 100$ and $\phi = 0.5$. We report $\phi = 0.5$ to illustrate behavior at an intermediate point along the trade-off. We present our results for $n = 200, 500$ and for different choices of GSW parameter $\phi$ in Appendix B. The robustness column measures $\|\text{Cov}(\boldsymbol{z})\|$. Each covariate moment column measures $\|\text{Cov}((\boldsymbol{X}^{\otimes k})^\top \boldsymbol{z})\|$ for $k = 1, \ldots, 5$. For all metrics, smaller values are better.

Across both sample sizes, GSW designs improve covariate balance relative to classical baselines. Bernoulli, CR, and blocking exhibit substantial covariate imbalance (e.g., for $n = 100$, first-moment ratios are above 11); matching remains far from optimal (e.g., ratio 5.15 on moment 1); Rerandomization performs better, but still leaves noticeable gaps in higher-order moments. GSW(1) from Harshaw et al. (2024) achieves the minimum first-moment imbalance for both $n = 30$ and $n = 100$, but it degrades on higher-order moments (e.g., moment 2 is 1.93 for $n = 30$ and 3.46 for $n = 100$, and moment 4 is 1.38 and 2.05, respectively). In contrast, our extensions GSW(2), GSW(3), and GSW($\infty$) consistently attain the best (or tied-best) performance on moments 2-5, with only a modest increase on moment 1 (e.g., GSW(2) has ratio 1.54 for $n = 30$ and 1.62 for $n = 100$). All GSW variants have a relatively good robustness with ratios at most 1.25.

**MSE.** Table 1 reports normalized MSE ratios. We use $\phi = 0.01$ for the MSE comparison because this setting emphasizes covariate balance and gives the strongest estimation gains in the low-noise regimes studied here. We present results for different values of $\phi$ in Appendix B.

Under *linear* outcomes, GSW(1) is optimal for all $n$, and it yields very large gains over classical baselines (e.g., at $n = 500$, Bernoulli/CR are over $100\times$ worse than the best design). For *nonlinear* outcomes, however, the ranking shifts: in the *quadratic* setting GSW(1) performs poorly (ratios around 2–5), while our higher-order variants achieve the best performance at $n = 30$ (GSW($\infty$)) and $n = 100$ (GSW(2)), and remain competitive as matching becomes optimal for larger $n$. For *cubic* outcomes, GSW designs consistently dominate the baselines, with GSW(3) best at $n = 100$ and GSW(1) best at the remaining sample sizes. For *sinusoidal* outcomes, GSW($\infty$) is best for $n \geq 100$ and near-optimal at $n = 30$, whereas GSW(1) again lags substantially behind. Overall, these results position GSW(2), GSW(3), and GSW($\infty$) as complementary extensions to the Harshaw et al. (2024) baseline: while GSW(1) excels when

*Table 1.* MSE ratios relative to the best method for $d = 10$ raw covariates. For each outcome type, MSEs are divided by the minimum MSE. Bern = Bernoulli; CR = complete randomization; Block = blocking; Match = pairwise matching; RR = rerandomization. All GSW variants have $\phi = 0.01$.

| | Linear outcomes ($\downarrow$) | | | | | | | | | Quadratic outcomes ($\downarrow$) | | | | | | | | |
|---|---|---|---|---|---|---|---|---|---|---|---|---|---|---|---|---|---|---|
| $n$ | Bern | CR | Block | Match | RR | GSW(1) | GSW(2) | GSW(3) | GSW($\infty$) | Bern | CR | Block | Match | RR | GSW(1) | GSW(2) | GSW(3) | GSW($\infty$) |
| 30 | 8.67 | 8.61 | 8.39 | 4.47 | 2.12 | **1.00** | 1.79 | 2.32 | 2.25 | 2.02 | 1.20 | 1.19 | 1.04 | 1.22 | 2.03 | 1.14 | 1.37 | **1.00** |
| 100 | 27.42 | 27.49 | 26.05 | 9.16 | 5.96 | **1.00** | 1.78 | 2.36 | 2.25 | 5.21 | 3.14 | 3.05 | 1.87 | 3.13 | 5.27 | **1.00** | 1.79 | 1.20 |
| 200 | 52.35 | 52.04 | 46.25 | 14.72 | 10.95 | **1.00** | 3.08 | 2.64 | 3.74 | 3.17 | 1.89 | 1.82 | **1.00** | 1.90 | 3.18 | 1.74 | 1.58 | 1.56 |
| 500 | 113.16 | 113.75 | 87.20 | 24.78 | 23.84 | **1.00** | 4.96 | 3.22 | 5.55 | 3.97 | 2.34 | 2.05 | **1.00** | 2.35 | 3.97 | 2.16 | 1.86 | 1.95 |

| | Cubic outcomes ($\downarrow$) | | | | | | | | | Sinusoidal outcomes ($\downarrow$) | | | | | | | | |
|---|---|---|---|---|---|---|---|---|---|---|---|---|---|---|---|---|---|---|
| $n$ | Bern | CR | Block | Match | RR | GSW(1) | GSW(2) | GSW(3) | GSW($\infty$) | Bern | CR | Block | Match | RR | GSW(1) | GSW(2) | GSW(3) | GSW($\infty$) |
| 30 | 2.56 | 2.54 | 2.48 | 1.75 | 1.23 | **1.00** | 1.17 | 1.23 | 1.24 | 4.93 | 2.99 | 2.94 | 1.61 | **1.00** | 2.75 | 1.51 | 1.95 | 1.08 |
| 100 | 3.32 | 3.31 | 3.23 | 1.55 | 1.51 | 1.09 | 1.18 | **1.00** | 1.08 | 12.18 | 7.09 | 6.83 | 2.75 | 2.22 | 6.16 | 1.25 | 1.80 | **1.00** |
| 200 | 3.14 | 3.13 | 2.92 | 1.32 | 1.41 | **1.00** | 1.10 | 1.02 | 1.11 | 10.38 | 6.07 | 5.42 | 1.93 | 1.85 | 5.19 | 1.65 | 1.48 | **1.00** |
| 500 | 3.28 | 3.30 | 2.73 | 1.11 | 1.47 | **1.00** | 1.09 | 1.02 | 1.11 | 11.88 | 6.94 | 5.41 | 1.74 | 2.08 | 5.75 | 1.66 | 1.31 | **1.00** |

*Table 2.* Robustness and covariate moment imbalance ratios to the best design for $d = 10$ (so the minimum equals 1). For each column metric, numbers are divided by the minimum number. Bern = Bernoulli; CR = complete randomization; Block = blocking; Match = pairwise matching; RR = rerandomization. All GSW variants have $\phi = 0.5$.

| | | | Covariate Moment ($\downarrow$) | | | | |
|---|---|---|---|---|---|---|---|
| $n$ | Design | Robustness ($\downarrow$) | 1 | 2 | 3 | 4 | 5 |
| 30 | Bern | **1.00** | 4.95 | 1.80 | 1.62 | 1.20 | 1.10 |
| | CR | 1.03 | 5.00 | 1.14 | 1.66 | 1.12 | 1.13 |
| | Block | 1.21 | 4.87 | 1.12 | 1.63 | 1.14 | 1.16 |
| | Match | 1.82 | 3.76 | 1.39 | 1.86 | 1.58 | 1.61 |
| | RR | 1.48 | 1.20 | 1.36 | 1.21 | 1.40 | 1.25 |
| | GSW(1) | 1.24 | **1.00** | 1.93 | 1.10 | 1.38 | 1.09 |
| | GSW(2) | 1.25 | 1.54 | 1.13 | 1.12 | **1.00** | 1.11 |
| | GSW(3) | 1.20 | 1.79 | 1.28 | **1.00** | 1.06 | **1.00** |
| | GSW($\infty$) | 1.18 | 2.10 | **1.00** | 1.08 | 1.05 | 1.04 |
| 100 | Bern | **1.00** | 12.12 | 3.42 | 2.36 | 1.99 | 1.36 |
| | CR | 1.01 | 12.18 | 1.52 | 2.38 | 1.56 | 1.37 |
| | Block | 1.37 | 11.42 | 1.49 | 2.26 | 1.58 | 1.38 |
| | Match | 1.72 | 5.15 | 1.27 | 1.62 | 1.64 | 1.48 |
| | RR | 1.07 | 2.60 | 1.57 | 1.17 | 1.62 | 1.11 |
| | GSW(1) | 1.07 | **1.00** | 3.46 | 1.15 | 2.05 | 1.10 |
| | GSW(2) | 1.17 | 1.62 | **1.00** | 1.23 | **1.00** | 1.20 |
| | GSW(3) | 1.15 | 1.99 | 1.28 | **1.00** | 1.12 | **1.00** |
| | GSW($\infty$) | 1.12 | 2.45 | 1.02 | 1.14 | 1.13 | 1.10 |

outcomes are linear, incorporating higher-order structure yields markedly improved performance under nonlinear response functions, which often cuts MSE by multiple-fold in the quadratic and sinusoidal settings and preserves large gains over standard randomization schemes across all $n$.

## Impact Statement

This paper presents work whose goal is to advance the field of Machine Learning. There are many potential societal consequences of our work, none which we feel must be specifically highlighted here.

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

# A. Missing Proofs

In this section, we include missing proofs from the main text.

## A.1. Missing Proofs in Section 3

*Proof of Lemma 3.2.* In Gram GSW, at each iteration $t$ the DIRECTIONORACLE solves a linear system of the form

$$\boldsymbol{u}_t(\mathcal{A}_t \setminus p_t) = -(\lambda \boldsymbol{I} + \boldsymbol{K}_t)^{-1} \boldsymbol{k}_{p_t},$$

where $\boldsymbol{K}_t$ is the principal submatrix of the Gram matrix $\boldsymbol{K}$ indexed by the current active set (excluding the pivot), and $\boldsymbol{k}_{p_t}$ is the corresponding restricted pivot column.

Precompute and store $\boldsymbol{K} \in \mathbb{R}^{n \times n}$, which uses $O(n^2)$ space. Maintain a Cholesky factorization

$$\lambda \boldsymbol{I} + \boldsymbol{K}_t = \boldsymbol{L}_t \boldsymbol{L}_t^\top$$

for the current principal submatrix. Solving for $\boldsymbol{u}_t$ then requires two triangular solves, costing $O(|\mathcal{A}_t|^2)$ time. Moreover, the step-size rule freezes at least one coordinate per iteration, so the active set shrinks by (at least) one each step; in the standard GSW walk it shrinks by exactly one, hence $|\mathcal{A}_t| = n - t + 1$. Updating $\boldsymbol{L}_t$ to the next iteration corresponds to deleting a row/column (passing to a smaller principal submatrix), which can be done via standard Cholesky downdates in $O(|\mathcal{A}_t|^2)$ time.

Therefore, the total runtime is

$$\sum_{t=1}^{n} O(|\mathcal{A}_t|^2) = \sum_{s=1}^{n} O(s^2) = O(n^3),$$

and the memory is dominated by storing $\boldsymbol{K}$ (and one Cholesky factor), i.e., $O(n^2)$. □

## A.2. Missing Proofs in Section 4

*Proof of Claim 4.1.* Note that

$$\|\psi_{\boldsymbol{\alpha}}(\boldsymbol{x})\|_{\mathcal{H}}^2 = \alpha_0 + \sum_{k \geq 1} \frac{\alpha_k}{\xi^{2k}} \|\boldsymbol{x}^{\otimes k}\|_{\mathcal{H}_k}^2 = \alpha_0 + \sum_{k \geq 1} \frac{\alpha_k}{\xi^{2k}} \|\boldsymbol{x}\|_2^{2k} \leq \sum_{k \geq 0} \alpha_k \leq 1.$$

□

*Proof of Lemma 4.2.* The expectation and covariance follow Lemma 3.1 and the GSW results in Harshaw et al. (2024).

The 1-subgaussianity does not immediately follow since $\boldsymbol{B}$ has an infinite number of rows. Let $\boldsymbol{R}^{n \times n}$ be a factor with $\boldsymbol{R}\boldsymbol{R}^\top = \boldsymbol{B}^* \boldsymbol{B}$. By Lemma 3.1, the distribution of $\boldsymbol{z}$ is the same as the distribution of the assignment vector returned by Feature GSW on the augmented covariate matrix with $\boldsymbol{R}$. Let $\boldsymbol{r}_i$ be the $i$th column of $\boldsymbol{R}$, and $\boldsymbol{b}_i$ be the $i$th column of $\boldsymbol{B}$. For any $\boldsymbol{\theta} \in \text{span}\{\boldsymbol{b}_i\}$, we can write $\boldsymbol{\theta} = \boldsymbol{B}\boldsymbol{y}$ for some $\boldsymbol{y} \in \mathbb{R}^n$, and

$$\langle \boldsymbol{\theta}, \boldsymbol{B}\boldsymbol{z} \rangle_{\mathcal{H}} = \langle \boldsymbol{y}, \boldsymbol{B}^* \boldsymbol{B}\boldsymbol{z} \rangle = \langle \boldsymbol{y}, \boldsymbol{R}\boldsymbol{R}^\top \boldsymbol{z} \rangle = \langle \boldsymbol{R}^\top \boldsymbol{y}, \boldsymbol{R}^\top \boldsymbol{z} \rangle,$$

which is $\|\boldsymbol{R}^\top \boldsymbol{y}\|_2^2$-subgaussian. For $\boldsymbol{\theta} \notin \text{span}\{\boldsymbol{b}_i\}$, let $\boldsymbol{\Pi}\boldsymbol{\theta}$ be its orthogonal projection onto $\text{span}\{\boldsymbol{b}_i\}$, and we apply the above argument to $\boldsymbol{\Pi}\boldsymbol{\theta}$. In addition, since $\|\boldsymbol{R}^\top \boldsymbol{y}\|_2 = \|\boldsymbol{B}\boldsymbol{y}\|_{\mathcal{H}} = \|\boldsymbol{\theta}\|_{\mathcal{H}}$, we have that $\boldsymbol{B}\boldsymbol{z}$ is 1-subgaussian. □

*Proof of Lemma 4.3.* We can write

$$f(\boldsymbol{x}) = \sum_{k=0}^{m} \frac{1}{k!} D^k f(\boldsymbol{0})[\boldsymbol{x}^{\otimes k}] + R_m(\boldsymbol{x}),$$

where $D^k f(\boldsymbol{0})$ is a symmetric $k$-linear form and $R_m(\boldsymbol{x})$ is the remainder:

$$|R_m(\boldsymbol{x})| \leq \frac{1}{(m+1)!} \sup_{t \in [0,1]} \left\| D^{m+1} f(t\boldsymbol{x}) \right\|_{\text{op}} \|\boldsymbol{x}\|_2^{m+1}.$$

Then,

$$S_f = \sum_{i=1}^{n} z_i f(\boldsymbol{x}_i) = \sum_{k=0}^{m} \frac{1}{k!} \langle D^k f(\boldsymbol{0}), \sum_{i=1}^{n} z_i \boldsymbol{x}_i^{\otimes k} \rangle_{\mathcal{H}_k} + \sum_{i=1}^{n} z_i R_m(\boldsymbol{x}_i).$$

By Lemma 4.2, $S_f$ is $(\frac{T_1}{1-\phi} + \frac{T_2}{\phi})^{1/2}$-subgaussian. $\square$

### A.3. Missing Proofs in Section 5

Proposition 5.1 follows Harshaw et al. (2024). We add a short proof for completeness.

*Proof of Proposition 5.1.* Let

$$\boldsymbol{B}\boldsymbol{z} = \left( \sqrt{\phi}\,\boldsymbol{z}, \sqrt{1-\phi}\,\boldsymbol{\Psi}^* \boldsymbol{z} \right) \in \mathbb{R}^n \oplus \mathcal{H}.$$

By Lemma 4.2, $\boldsymbol{B}\boldsymbol{z}$ is subgaussian with variance proxy 1, and hence for every $\boldsymbol{\theta} \in \mathbb{R}^n \oplus \mathcal{H}$, the random variable $\langle \boldsymbol{\theta}, \boldsymbol{B}\boldsymbol{z} \rangle$ is subgaussian with variance proxy $\|\boldsymbol{\theta}\|^2$.

For any $\boldsymbol{\beta} \in \mathcal{H}$, define

$$\boldsymbol{\theta}_\beta = \left( \frac{\boldsymbol{\mu} - \boldsymbol{\Psi}\boldsymbol{\beta}}{\sqrt{\phi}}, \frac{\boldsymbol{\beta}}{\sqrt{1-\phi}} \right) \in \mathbb{R}^n \oplus \mathcal{H}.$$

Then

$$\begin{aligned}
\langle \boldsymbol{\theta}_\beta, \boldsymbol{B}\boldsymbol{z} \rangle &= \left\langle \frac{\boldsymbol{\mu} - \boldsymbol{\Psi}\boldsymbol{\beta}}{\sqrt{\phi}}, \sqrt{\phi}\,\boldsymbol{z} \right\rangle + \left\langle \frac{\boldsymbol{\beta}}{\sqrt{1-\phi}}, \sqrt{1-\phi}\,\boldsymbol{\Psi}^* \boldsymbol{z} \right\rangle_{\mathcal{H}} \\
&= (\boldsymbol{\mu} - \boldsymbol{\Psi}\boldsymbol{\beta})^\top \boldsymbol{z} + \langle \boldsymbol{\beta}, \boldsymbol{\Psi}^* \boldsymbol{z} \rangle_{\mathcal{H}} \\
&= (\boldsymbol{\mu} - \boldsymbol{\Psi}\boldsymbol{\beta})^\top \boldsymbol{z} + (\boldsymbol{\Psi}\boldsymbol{\beta})^\top \boldsymbol{z} \\
&= \boldsymbol{\mu}^\top \boldsymbol{z}.
\end{aligned}$$

Therefore, for every $\boldsymbol{\beta} \in \mathcal{H}$, the random variable $\boldsymbol{\mu}^\top \boldsymbol{z}$ is subgaussian with variance proxy

$$\|\boldsymbol{\theta}_\beta\|^2 = \frac{1}{\phi} \|\boldsymbol{\mu} - \boldsymbol{\Psi}\boldsymbol{\beta}\|_2^2 + \frac{1}{1-\phi} \|\boldsymbol{\beta}\|_{\mathcal{H}}^2.$$

Since the Horvitz-Thompson estimator satisfies

$$\hat{\tau} - \tau = \frac{1}{n} \boldsymbol{\mu}^\top \boldsymbol{z},$$

it follows that, for every $\boldsymbol{\beta} \in \mathcal{H}$, $\hat{\tau} - \tau$ is subgaussian with variance proxy

$$\frac{1}{n^2} \left( \frac{1}{\phi} \|\boldsymbol{\mu} - \boldsymbol{\Psi}\boldsymbol{\beta}\|_2^2 + \frac{1}{1-\phi} \|\boldsymbol{\beta}\|_{\mathcal{H}}^2 \right).$$

Taking the infimum over $\boldsymbol{\beta} \in \mathcal{H}$ gives the variance proxy $L_{\phi,\alpha}/n$. Hence

$$\mathrm{var}(\hat{\tau}) \leq \frac{L_{\phi,\alpha}}{n}.$$

The same subgaussian bound gives, for any $\gamma > 0$,

$$\mathbb{P}\left( |\hat{\tau} - \tau| \geq \gamma \right) \leq 2 \exp\left( -\frac{\gamma^2}{2L_{\phi,\alpha}/n} \right) = 2 \exp\left( -\frac{\gamma^2 n}{2L_{\phi,\alpha}} \right).$$

This proves the result. $\square$

*Proof of Corollary 5.2.* For any $\boldsymbol{\beta}_1 \in \mathbb{R}^d$, let $f(\boldsymbol{\beta}_1) : \mathbb{R}^d \to \mathcal{H}$ defined as

$$f(\boldsymbol{\beta}_1) \mapsto \begin{pmatrix} 0 \\ (\sqrt{\alpha_1})^{-1}\xi\boldsymbol{\beta}_1 \\ \mathbf{0} \end{pmatrix}.$$

Then,

$$L_{\phi,\alpha} \leq \frac{1}{\phi n} \left\| \boldsymbol{\mu} - \boldsymbol{\Psi} f(\boldsymbol{\beta}_1) \right\|^2 + \frac{1}{(1-\phi)n} \left\| f(\boldsymbol{\beta}_1) \right\|^2$$

$$= \frac{1}{\phi n} \left\| \boldsymbol{\mu} - \boldsymbol{X}\boldsymbol{\beta}_1 \right\|^2 + \frac{\xi^2}{(1-\phi)n\,\alpha_1} \left\| \boldsymbol{\beta}_1 \right\|^2.$$

Since the above inequality holds for any $\boldsymbol{\beta}_1 \in \mathbb{R}^d$, we have

$$L_{\phi,\alpha} \leq \frac{1}{\alpha_1} \min_{\boldsymbol{\beta}_1 \in \mathbb{R}^d} \left( \frac{1}{\phi n} \left\| \boldsymbol{\mu} - \boldsymbol{X}\boldsymbol{\beta}_1 \right\|^2 + \frac{\xi^2}{(1-\phi)n} \left\| \boldsymbol{\beta}_1 \right\|^2 \right).$$

$\square$

*Proof of Lemma 5.3.* For the two-moment feature map, the normalized feature matrix is

$$\boldsymbol{\Psi}_\alpha = \begin{pmatrix} \sqrt{\alpha}\,\xi^{-1}\boldsymbol{X} & \sqrt{1-\alpha}\,\xi^{-2}\boldsymbol{X}^{\otimes 2} \end{pmatrix}.$$

Thus

$$\boldsymbol{\Psi}_\alpha \boldsymbol{\Psi}_\alpha^\top = \alpha\xi^{-2}\boldsymbol{K}_1 + (1-\alpha)\xi^{-4}\boldsymbol{K}_2.$$

By the ridge formula in Equation (6) in Proposition 5.1,

$$L_{\phi,\alpha} = \frac{1}{(1-\phi)n}\boldsymbol{\mu}^\top \left( \frac{\phi}{1-\phi}\boldsymbol{I} + \boldsymbol{\Psi}_\alpha \boldsymbol{\Psi}_\alpha^\top \right)^{-1}\boldsymbol{\mu}$$

$$= \frac{\xi^2}{(1-\phi)n}\boldsymbol{\mu}^\top \boldsymbol{A}(\alpha)^{-1}\boldsymbol{\mu},$$

where

$$\boldsymbol{A}(\alpha) = \lambda\boldsymbol{I} + \alpha\boldsymbol{K}_1 + (1-\alpha)\xi^{-2}\boldsymbol{K}_2, \qquad \lambda = \frac{\phi\xi^2}{1-\phi}.$$

Let

$$\boldsymbol{D} \stackrel{\text{def}}{=} \boldsymbol{K}_1 - \xi^{-2}\boldsymbol{K}_2, \qquad c \stackrel{\text{def}}{=} \frac{\xi^2}{(1-\phi)n}.$$

Then $\boldsymbol{A}'(\alpha) = \boldsymbol{D}$, and

$$\frac{d}{d\alpha}\boldsymbol{A}(\alpha)^{-1} = -\boldsymbol{A}(\alpha)^{-1}\boldsymbol{D}\boldsymbol{A}(\alpha)^{-1}.$$

Therefore,

$$\frac{d}{d\alpha}L_{\phi,\alpha} = -c\,\boldsymbol{\mu}^\top \boldsymbol{A}(\alpha)^{-1}\boldsymbol{D}\boldsymbol{A}(\alpha)^{-1}\boldsymbol{\mu} = -c\,\boldsymbol{w}(\alpha)^\top \boldsymbol{D}\boldsymbol{w}(\alpha) = -c\,g(\alpha).$$

Also,

$$\frac{d^2}{d\alpha^2}L_{\phi,\alpha} = 2c\,\boldsymbol{w}(\alpha)^\top \boldsymbol{D}\boldsymbol{A}(\alpha)^{-1}\boldsymbol{D}\boldsymbol{w}(\alpha) \geq 0,$$

since $A(\alpha) \succ 0$. Hence $L_{\phi,\alpha}$ is convex in $\alpha$.

If $g(1) < 0$, then $L'_{\phi,\alpha}|_{\alpha=1} > 0$. Hence moving slightly left from $\alpha = 1$ decreases $L_{\phi,\alpha}$, so $\alpha = 1$ cannot be a minimizer. Therefore any minimizer satisfies $\alpha^\star < 1$.

Similarly, if $g(0) > 0$, then $L'_{\phi,\alpha}|_{\alpha=0} < 0$. Hence moving slightly right from $\alpha = 0$ decreases $L_{\phi,\alpha}$, so $\alpha = 0$ cannot be a minimizer. Therefore any minimizer satisfies $\alpha^\star > 0$.

If both $g(1) < 0$ and $g(0) > 0$ hold, then every minimizer is interior. Since $L_{\phi,\alpha}$ is differentiable, every interior minimizer satisfies

$$0 = L'_{\phi,\alpha}|_{\alpha=\alpha^\star} = -c\,g(\alpha^\star),$$

and hence $g(\alpha^\star) = 0$.

It remains to justify the uniqueness of the root. Since $L_{\phi,\alpha}$ is convex, $L'_{\phi,\alpha}$ is nondecreasing. Therefore $g$ is nonincreasing. If $g$ had two distinct zeros, then $L'_{\phi,\alpha}$ would be zero on the interval between them. But $L'_{\phi,\alpha}$ is analytic in $\alpha$, because $\boldsymbol{A}(\alpha) \succ 0$ for all $\alpha \in [0,1]$. Hence $L'_{\phi,\alpha}$ would be identically zero on $[0,1]$, contradicting $g(0) > 0$ and $g(1) < 0$. Thus $g(\alpha) = 0$ has a unique solution in $(0,1)$. $\qquad\square$

### A.4. Missing Proofs in Section 6

*Proof of Lemma 6.1.* Since $\tilde{K}$ is symmetric PSD, by Lemma 4.2, we have

$$\mathbb{E}[\tilde{z}] = \boldsymbol{0}, \qquad \operatorname{Cov}(\tilde{z}) \preccurlyeq \left(\phi \boldsymbol{I} + (1-\phi)\tilde{\boldsymbol{K}}\right)^{-1}.$$

Since $\left\|\boldsymbol{K} - \tilde{\boldsymbol{K}}\right\| \le \epsilon_k < \frac{\phi}{1-\phi}$, we have

$$\operatorname{Cov}(\tilde{z}) \preccurlyeq \left(\phi \boldsymbol{I} + (1-\phi)\boldsymbol{K} + (1-\phi)(\tilde{\boldsymbol{K}} - \boldsymbol{K})\right)^{-1}$$
$$\preccurlyeq \left((\phi - (1-\phi)\epsilon_k)\boldsymbol{I} + (1-\phi)\boldsymbol{K}\right)^{-1}.$$

$\qquad\square$

Lemma 6.2 follows Lemma 6.1 and the standard ridge regression formula.

## B. Additional Experimental Details and Results

### B.1. Outcome Generating Process

We provide additional details about the potential outcomes for randomly generated covariates.

For each unit $i \in [n]$, the potential outcome $\mu_i = a_i + b_i$ (the sum of two potential outcomes) is generated by

$$\mu_i = f(\boldsymbol{x}_i) + \epsilon_i, \quad \text{where } \epsilon_i \sim \mathcal{N}(0, \sigma^2).$$

By default, we set $\sigma = 0.1$. The value of $\sigma$ controls the extent to which the outcomes are uncorrelated with the covariates. We also consider different values of $\sigma$.

We consider four choices of the function $f$, each depending only on the first five covariates, similar to Kallus (2018).

$$s(\boldsymbol{x}) = \boldsymbol{x}(1) + 2\boldsymbol{x}(2) + 3\boldsymbol{x}(3) + 4\boldsymbol{x}(4) + 5\boldsymbol{x}(5).$$

The four functions are

1. Linear: $f(\boldsymbol{x}) = s(\boldsymbol{x})$;

2. Quadratic: $f(\boldsymbol{x}) = s(\boldsymbol{x})^2$;

3. Cubic: $f(\boldsymbol{x}) = s(\boldsymbol{x})^3$;

4. Sinusoidal: $f(\boldsymbol{x}) = \sin(\pi/3 + \pi\boldsymbol{x}(1)/3 - 2\pi\boldsymbol{x}(2)/3) - 6\sin(\pi\boldsymbol{x}(1)/3 + \pi\boldsymbol{x}(2)/4) + 6\sin(\pi\boldsymbol{x}(1)/3 + \pi\boldsymbol{x}(2)/6)$.

*Table 3.* Robustness and covariate moment imbalance ratios to the best design for $d = 10$ and $n = 30$, full table. For each column metric, numbers are divided by the minimum number. CR = complete randomization; Block = blocking; Match = pairwise matching; RR = rerandomization; LR = low rank. For GSW variant, the last parameter is the value of $\phi$.

| $n$ | Design | Robustness ($\downarrow$) | Covariate Moment ($\downarrow$) 1 | 2 | 3 | 4 | 5 |
|---|---|---|---|---|---|---|---|
| 30 | Bernoulli | **1.00** | 12.42 | 2.53 | 1.62 | 1.20 | 1.13 |
| | CR | 1.03 | 12.55 | 1.60 | 1.66 | 1.12 | 1.16 |
| | Block | 1.22 | 12.21 | 1.58 | 1.63 | 1.14 | 1.19 |
| | Match | 1.82 | 9.44 | 1.95 | 1.86 | 1.58 | 1.65 |
| | RR | 1.48 | 3.02 | 1.90 | 1.21 | 1.40 | 1.28 |
| | GSW($\infty$) LR 0.01 | 1.28 | 3.10 | 1.49 | 1.15 | 1.12 | 1.17 |
| | GSW($\infty$) LR 0.1 | 1.27 | 3.04 | 1.48 | 1.12 | 1.12 | 1.14 |
| | GSW($\infty$) LR 0.5 | 1.12 | 5.79 | 1.53 | 1.09 | 1.11 | 1.04 |
| | GSW($\infty$) LR 0.9 | 1.03 | 10.68 | 1.58 | 1.44 | 1.10 | 1.06 |
| | GSW($\infty$) LR 0.99 | 1.01 | 12.18 | 2.32 | 1.60 | 1.16 | 1.12 |
| | GSW(3) LR 0.01 | 1.72 | 2.18 | 1.25 | 1.13 | 1.14 | 1.29 |
| | GSW(3) LR 0.1 | 1.55 | 2.47 | 1.34 | 1.08 | 1.09 | 1.20 |
| | GSW(3) LR 0.5 | 1.20 | 4.48 | 1.82 | **1.00** | 1.06 | 1.02 |
| | GSW(3) LR 0.9 | 1.03 | 9.59 | 2.35 | 1.30 | 1.15 | 1.01 |
| | GSW(3) LR 0.99 | **1.00** | 12.01 | 2.53 | 1.58 | 1.21 | 1.11 |
| | GSW(2) LR 0.01 | 2.06 | 1.70 | 1.03 | 1.72 | 1.26 | 1.83 |
| | GSW(2) LR 0.1 | 1.77 | 1.74 | 1.06 | 1.51 | 1.15 | 1.60 |
| | GSW(2) LR 0.5 | 1.24 | 3.66 | 1.61 | 1.12 | 1.02 | 1.13 |
| | GSW(2) LR 0.9 | 1.03 | 9.22 | 2.33 | 1.28 | 1.14 | 1.02 |
| | GSW(2) LR 0.99 | 1.01 | 12.09 | 2.52 | 1.58 | 1.20 | 1.11 |
| | GSW(1) LR 0.01 | 1.43 | **1.00** | 2.82 | 1.22 | 1.48 | 1.27 |
| | GSW(1) LR 0.1 | 1.40 | 1.09 | 2.80 | 1.20 | 1.46 | 1.24 |
| | GSW(1) LR 0.5 | 1.24 | 2.51 | 2.72 | 1.09 | 1.38 | 1.11 |
| | GSW(1) LR 0.9 | 1.05 | 8.04 | 2.56 | 1.17 | 1.23 | **1.00** |
| | GSW(1) LR 0.99 | 1.01 | 11.85 | 2.53 | 1.56 | 1.20 | 1.10 |
| | GSW($\infty$) 0.01 | 1.99 | 2.39 | 1.09 | 1.46 | 1.21 | 1.62 |
| | GSW($\infty$) 0.1 | 1.61 | 2.73 | 1.16 | 1.28 | 1.08 | 1.37 |
| | GSW($\infty$) 0.5 | 1.18 | 5.27 | 1.41 | 1.08 | 1.05 | 1.07 |
| | GSW($\infty$) 0.9 | 1.03 | 10.28 | 1.56 | 1.40 | 1.09 | 1.05 |
| | GSW($\infty$) 0.99 | **1.00** | 12.07 | 2.31 | 1.59 | 1.15 | 1.11 |
| | GSW(3) 0.01 | 1.76 | 2.26 | 1.26 | 1.14 | 1.16 | 1.31 |
| | GSW(3) 0.1 | 1.56 | 2.52 | 1.33 | 1.09 | 1.10 | 1.21 |
| | GSW(3) 0.5 | 1.20 | 4.48 | 1.80 | **1.00** | 1.06 | 1.02 |
| | GSW(3) 0.9 | 1.02 | 9.45 | 2.37 | 1.30 | 1.15 | 1.01 |
| | GSW(3) 0.99 | **1.00** | 11.94 | 2.54 | 1.58 | 1.20 | 1.11 |
| | GSW(2) 0.01 | 2.78 | 1.83 | **1.00** | 2.07 | 1.58 | 2.32 |
| | GSW(2) 0.1 | 1.99 | 2.04 | 1.06 | 1.60 | 1.21 | 1.73 |
| | GSW(2) 0.5 | 1.26 | 3.86 | 1.59 | 1.12 | **1.00** | 1.14 |
| | GSW(2) 0.9 | 1.03 | 9.06 | 2.31 | 1.27 | 1.14 | 1.01 |
| | GSW(2) 0.99 | **1.00** | 11.88 | 2.53 | 1.57 | 1.20 | 1.10 |
| | GSW(1) 0.01 | 1.66 | 1.11 | 2.92 | 1.35 | 1.60 | 1.44 |
| | GSW(1) 0.1 | 1.51 | 1.24 | 2.86 | 1.26 | 1.52 | 1.32 |
| | GSW(1) 0.5 | 1.24 | 2.52 | 2.72 | 1.10 | 1.38 | 1.12 |
| | GSW(1) 0.9 | 1.04 | 7.95 | 2.57 | 1.17 | 1.24 | **1.00** |
| | GSW(1) 0.99 | **1.00** | 11.68 | 2.57 | 1.55 | 1.21 | 1.09 |

## B.2. Robustness and Covariate Balance

In Tables 3 – 7, we report robustness and covariate moment imbalance ratios over $\phi = 0.01, 0.1, 0.5, 0.9, 0.99$ for GSW and sample sizes $n = 30, 100, 200, 500$ and $d = 10, 30$.

Classical randomization schemes exhibit substantial covariate imbalance, particularly in the first moment, and the gap to the best design increases with $n$. Harshaw et al. (2024) GSW(1) consistently attains the best first-moment balance but can leave large higher-order moment imbalance. In contrast, our nonlinear extensions GSW(2), GSW(3), and GSW($\infty$) systematically improve higher-order moments (often achieving the minimum for moments 2–5) with only modest robustness degradation. The parameter $\phi$ provides an explicit knob that trades balance for randomness/robustness; small $\phi$ yields strong higher-moment balance, while $\phi \to 1$ approaches fully randomized designs.

**Low rank implementation.** For $n = 30, 100$, we test both GSW (non-LR) and its low-rank implementation (LR). The corresponding LR vs non-LR entries are close enough to credibly claim the LR approximation preserves performance rankings. So, the scalable implementation does not change the qualitative conclusions.

## B.3. MSE

In Tables 8 – 11, we report MSE ratios.

*Table 4.* Robustness and covariate moment imbalance ratios to the best design for $d = 10$ and $n = 100$, full table. For each column metric, numbers are divided by the minimum number. CR = complete randomization; Block = blocking; Match = pairwise matching; RR = rerandomization; LR = low rank. For GSW variant, the last parameter is the value of $\phi$.

| $n$ | Design | Robustness ($\downarrow$) | Covariate Moment ($\downarrow$) | | | | |
|---|---|---|---|---|---|---|---|
| | | | 1 | 2 | 3 | 4 | 5 |
| 100 | Bernoulli | **1.00** | 35.21 | 9.48 | 2.90 | 2.04 | 1.36 |
| | CR | 1.01 | 35.39 | 4.22 | 2.92 | 1.59 | 1.37 |
| | Block | 1.37 | 33.19 | 4.14 | 2.78 | 1.62 | 1.38 |
| | Match | 1.72 | 14.97 | 3.53 | 1.99 | 1.68 | 1.48 |
| | RR | 1.07 | 7.56 | 4.34 | 1.43 | 1.66 | 1.11 |
| | GSW($\infty$) LR 0.01 | 1.06 | 8.01 | 3.86 | 1.43 | 1.50 | 1.10 |
| | GSW($\infty$) LR 0.1 | 1.06 | 8.39 | 3.88 | 1.43 | 1.50 | 1.10 |
| | GSW($\infty$) LR 0.5 | 1.04 | 12.97 | 4.01 | 1.49 | 1.54 | 1.09 |
| | GSW($\infty$) LR 0.9 | 1.01 | 26.59 | 4.17 | 2.26 | 1.58 | 1.18 |
| | GSW($\infty$) LR 0.99 | **1.00** | 34.32 | 7.11 | 2.83 | 1.68 | 1.33 |
| | GSW(3) LR 0.01 | 1.19 | 2.45 | 2.42 | 1.27 | 1.12 | 1.04 |
| | GSW(3) LR 0.1 | 1.18 | 2.34 | 2.33 | 1.26 | 1.10 | 1.03 |
| | GSW(3) LR 0.5 | 1.09 | 5.64 | 4.11 | 1.33 | 1.30 | 1.05 |
| | GSW(3) LR 0.9 | 1.02 | 20.14 | 7.84 | 1.78 | 1.74 | 1.09 |
| | GSW(3) LR 0.99 | **1.00** | 33.02 | 9.34 | 2.74 | 2.02 | 1.31 |
| | GSW(2) LR 0.01 | 1.14 | 2.19 | 2.42 | 1.52 | 1.11 | 1.18 |
| | GSW(2) LR 0.1 | 1.13 | 2.42 | 2.53 | 1.50 | 1.12 | 1.17 |
| | GSW(2) LR 0.5 | 1.08 | 5.38 | 3.89 | 1.44 | 1.29 | 1.12 |
| | GSW(2) LR 0.9 | 1.02 | 19.06 | 7.60 | 1.74 | 1.69 | 1.10 |
| | GSW(2) LR 0.99 | **1.00** | 32.81 | 9.22 | 2.72 | 1.99 | 1.29 |
| | GSW(1) LR 0.01 | 1.08 | **1.00** | 9.60 | 1.44 | 2.11 | 1.12 |
| | GSW(1) LR 0.1 | 1.08 | 1.12 | 9.60 | 1.44 | 2.11 | 1.12 |
| | GSW(1) LR 0.5 | 1.07 | 2.91 | 9.50 | 1.43 | 2.08 | 1.11 |
| | GSW(1) LR 0.9 | 1.03 | 14.24 | 9.53 | 1.51 | 2.06 | 1.09 |
| | GSW(1) LR 0.99 | **1.00** | 30.77 | 9.51 | 2.57 | 2.05 | 1.25 |
| | GSW($\infty$) 0.01 | 1.95 | 2.44 | 1.14 | 1.35 | 1.23 | 1.30 |
| | GSW($\infty$) 0.1 | 1.43 | 2.95 | 1.46 | 1.37 | **1.00** | 1.17 |
| | GSW($\infty$) 0.5 | 1.12 | 7.11 | 2.82 | 1.40 | 1.16 | 1.10 |
| | GSW($\infty$) 0.9 | 1.03 | 22.64 | 3.96 | 1.96 | 1.51 | 1.12 |
| | GSW($\infty$) 0.99 | 1.01 | 33.54 | 6.74 | 2.77 | 1.63 | 1.32 |
| | GSW(3) 0.01 | 2.22 | 2.41 | 1.54 | **1.00** | 1.44 | 1.21 |
| | GSW(3) 0.1 | 1.59 | 2.74 | 1.79 | 1.04 | 1.15 | 1.02 |
| | GSW(3) 0.5 | 1.15 | 5.77 | 3.55 | 1.23 | 1.15 | **1.00** |
| | GSW(3) 0.9 | 1.03 | 19.50 | 7.43 | 1.74 | 1.65 | 1.08 |
| | GSW(3) 0.99 | 1.01 | 32.78 | 9.11 | 2.72 | 1.96 | 1.30 |
| | GSW(2)0.01 | 2.69 | 1.82 | **1.00** | 2.71 | 1.65 | 2.37 |
| | GSW(2)0.1 | 1.65 | 2.13 | 1.20 | 1.95 | 1.10 | 1.60 |
| | GSW(2)0.5 | 1.17 | 4.70 | 2.77 | 1.51 | 1.02 | 1.20 |
| | GSW(2)0.9 | 1.03 | 17.94 | 6.94 | 1.66 | 1.57 | 1.10 |
| | GSW(2)0.99 | 1.01 | 32.30 | 9.04 | 2.68 | 1.95 | 1.29 |
| | GSW(1) 0.01 | 1.11 | 1.17 | 9.69 | 1.46 | 2.13 | 1.14 |
| | GSW(1) 0.1 | 1.10 | 1.33 | 9.66 | 1.46 | 2.12 | 1.13 |
| | GSW(1) 0.5 | 1.07 | 2.93 | 9.58 | 1.42 | 2.10 | 1.10 |
| | GSW(1) 0.9 | 1.03 | 14.18 | 9.43 | 1.50 | 2.04 | 1.09 |
| | GSW(1) 0.99 | 1.01 | 30.95 | 9.35 | 2.58 | 2.01 | 1.26 |

Tables 8 – 9 show that classical designs have significantly larger MSE, especially for linear outcomes. For linear outcomes, GSW(1) achieves the smallest MSE. For quadratic outcomes, pairwise matching is best, and for sinusoidal outcomes, the best performance is attained by GSW($\infty$) with small $\phi$. Across outcome classes, these tables highlight the benefit of tailoring the design to the response structure rather than relying on a single baseline randomization scheme.

Tables 10 – 11 confirm the same qualitative patterns under higher noise ($n = 200$, $\sigma = 0.5$) and higher dimension ($n = 200$, $d = 30$, $\sigma = 0.1$). With $\sigma = 0.5$, the spread in ratios compresses relative to $\sigma = 0.1$ (e.g., Bernoulli linear 16.68 instead of 52.35), but GSW(1) remains optimal for linear outcomes and matching remains optimal for quadratic outcomes. In the higher-dimensional setting ($d = 30$), the sinusoidal column further emphasizes the importance of higher-order balancing: GSW($\infty$) with small $\phi$ achieves the minimum, while designs optimized for linear structure can be far from optimal for nonlinear responses (e.g., GSW(1) at $\phi = 0.01$ yields sinusoidal ratio 11.97). Finally, within each GSW family, increasing $\phi$ consistently pushes performance toward fully randomized behavior, degrading ratios as $\phi \to 1$, illustrating the intended balance-randomness tradeoff controlled by $\phi$.

**Additional comparison with kernel rerandomization.**  We also compare against a kernel-discrepancy-based framework for Rerandomization introduced by Li et al. (2019). We compare with two kernels: a quadratic kernel and an $L_2$-discrepancy kernel. We use the same simulation setting as Section 7 with $n = 100$, $d = 10$, and $\sigma = 0.1$ in Table 12. The kernel-rerandomization variants provide natural tractable nonlinear baselines, but in this setting they are not competitive with the

*Table 5.* Robustness and covariate moment imbalance ratios to the best design for $d = 10$ and $n = 200$, full table. For each column metric, numbers are divided by the minimum number. CR = complete randomization; Block = blocking; Match = pairwise matching; RR = rerandomization; LR = low rank. For GSW variant, the last parameter is the value of $\phi$.

| | | | Covariate Moment ($\downarrow$) | | | | |
|---|---|---|---|---|---|---|---|
| $n$ | Design | Robustness ($\downarrow$) | 1 | 2 | 3 | 4 | 5 |
| 200 | Bernoulli | **1.00** | 66.06 | 4.81 | 2.72 | 2.30 | 1.50 |
| | CR | **1.00** | 65.98 | 1.78 | 2.72 | 1.56 | 1.51 |
| | Block | 1.34 | 59.27 | 1.72 | 2.49 | 1.56 | 1.46 |
| | Match | 1.65 | 21.94 | 1.20 | 1.35 | 1.32 | 1.28 |
| | RR | 1.03 | 13.83 | 1.82 | 1.14 | 1.59 | 1.10 |
| | GSW($\infty$) LR 0.01 | 1.06 | 3.93 | 1.19 | 1.13 | 1.13 | 1.10 |
| | GSW($\infty$) LR 0.1 | 1.05 | 4.53 | 1.24 | 1.13 | 1.16 | 1.10 |
| | GSW($\infty$) LR 0.5 | 1.03 | 10.94 | 1.50 | 1.13 | 1.34 | 1.09 |
| | GSW($\infty$) LR 0.9 | 1.01 | 36.90 | 1.75 | 1.62 | 1.53 | 1.14 |
| | GSW($\infty$) LR 0.99 | **1.00** | 61.17 | 2.82 | 2.54 | 1.60 | 1.42 |
| | GSW(3) LR 0.01 | 1.07 | 2.72 | **1.00** | **1.00** | **1.00** | **1.00** |
| | GSW(3) LR 0.1 | 1.07 | 2.90 | 1.02 | **1.00** | 1.01 | **1.00** |
| | GSW(3) LR 0.5 | 1.04 | 6.93 | 1.54 | 1.06 | 1.22 | 1.04 |
| | GSW(3) LR 0.9 | 1.01 | 28.95 | 3.57 | 1.34 | 1.77 | 1.09 |
| | GSW(3) LR 0.99 | **1.00** | 59.17 | 4.64 | 2.45 | 2.23 | 1.39 |
| | GSW(2) LR 0.01 | 1.06 | 3.27 | 1.14 | 1.16 | 1.09 | 1.12 |
| | GSW(2) LR 0.1 | 1.05 | 3.57 | 1.17 | 1.15 | 1.11 | 1.12 |
| | GSW(2) LR 0.5 | 1.04 | 7.11 | 1.52 | 1.14 | 1.25 | 1.10 |
| | GSW(2) LR 0.9 | 1.02 | 27.25 | 3.38 | 1.30 | 1.70 | 1.10 |
| | GSW(2) LR 0.99 | **1.00** | 57.53 | 4.62 | 2.40 | 2.22 | 1.37 |
| | GSW(1) LR 0.01 | 1.04 | **1.00** | 4.86 | 1.13 | 2.34 | 1.10 |
| | GSW(1) LR 0.1 | 1.04 | 1.13 | 4.84 | 1.13 | 2.33 | 1.09 |
| | GSW(1) LR 0.5 | 1.04 | 3.05 | 4.82 | 1.13 | 2.32 | 1.10 |
| | GSW(1) LR 0.9 | 1.02 | 17.94 | 4.76 | 1.16 | 2.29 | 1.10 |
| | GSW(1) LR 0.99 | **1.00** | 52.70 | 4.80 | 2.20 | 2.30 | 1.28 |

best GSW configurations.

## B.4. Sensitivity to $\phi$ and Feature Weights

We further examine how GSW(2) changes with the balance-robustness parameter $\phi$ and the relative weight placed on second-order balance. Figure 4 reports covariate moment imbalance over a grid of $\phi$ and $\alpha_2/\alpha_1$. The figure is intended as a representative sensitivity check rather than a full sweep over all data-generating settings. As expected, smaller $\phi$ places more emphasis on balance, while increasing $\alpha_2/\alpha_1$ improves second-moment balance at the cost of weaker first-moment balance. This supports the default choice of keeping $\alpha_1$ relatively large while assigning a positive, smaller weight to $\alpha_2$.

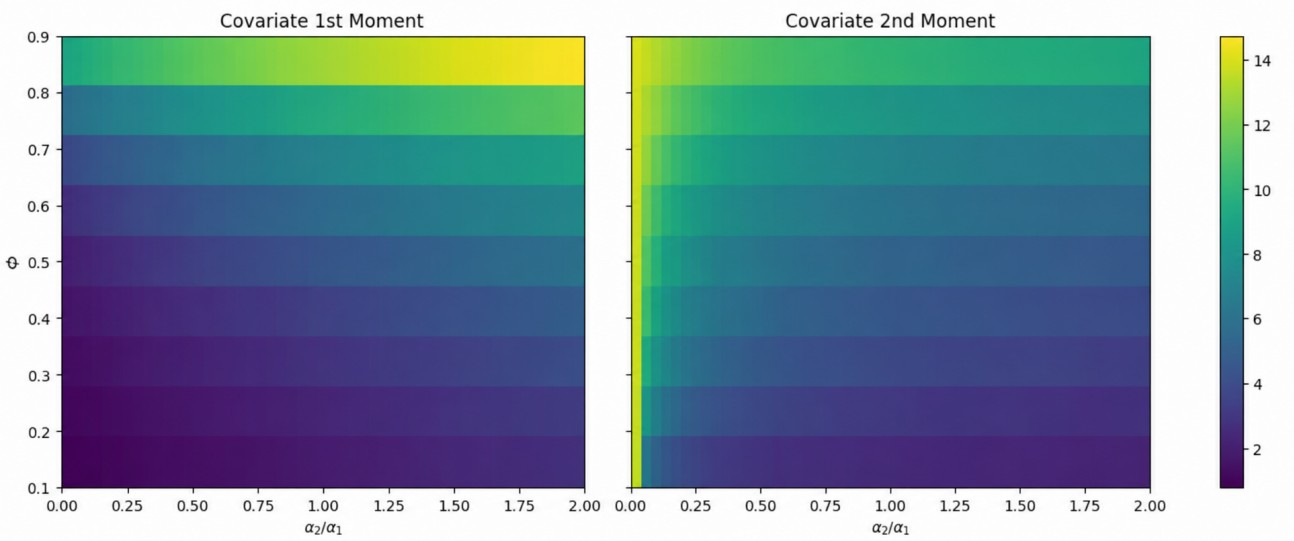

*Figure 4.* Sensitivity to balance-robustness parameter $\phi$ and feature weights. $n = 100, d = 10, \alpha_k = 0$ for $k \neq 1, 2$.

*Table 6.* Robustness and covariate moment imbalance ratios to the best design for $d = 10$ and $n = 500$, full table. For each column metric, numbers are divided by the minimum number. CR = complete randomization; Block = blocking; Match = pairwise matching; RR = rerandomization; LR = low rank. For GSW variant, the last parameter is the value of $\phi$.

| $n$ | Design | Robustness ($\downarrow$) | Covariate Moment ($\downarrow$) 1 | 2 | 3 | 4 | 5 |
|-----|--------|---------------------------|------|------|------|------|------|
| 500 | Bernoulli | **1.00** | 154.38 | 6.34 | 3.13 | 2.85 | 1.73 |
|     | CR | **1.00** | 155.25 | 1.98 | 3.15 | 1.62 | 1.74 |
|     | Block | 1.31 | 118.82 | 1.79 | 2.49 | 1.52 | 1.49 |
|     | Match | 1.54 | 37.82 | **1.00** | 1.03 | **1.00** | **1.00** |
|     | RR | 1.01 | 31.96 | 1.99 | 1.10 | 1.62 | 1.09 |
|     | GSW($\infty$) LR 0.01 | 1.02 | 6.95 | 1.43 | 1.07 | 1.22 | 1.08 |
|     | GSW($\infty$) LR 0.1 | 1.02 | 7.70 | 1.46 | 1.07 | 1.24 | 1.08 |
|     | GSW($\infty$) LR 0.5 | 1.01 | 15.57 | 1.63 | 1.07 | 1.36 | 1.08 |
|     | GSW($\infty$) LR 0.9 | 1.01 | 60.13 | 1.88 | 1.36 | 1.55 | 1.10 |
|     | GSW($\infty$) LR 0.99 | **1.00** | 132.61 | 2.35 | 2.71 | 1.61 | 1.53 |
|     | GSW(3) LR 0.01 | 1.02 | 3.85 | 1.18 | **1.00** | 1.04 | 1.02 |
|     | GSW(3) LR 0.1 | 1.02 | 4.22 | 1.21 | **1.00** | 1.06 | 1.02 |
|     | GSW(3) LR 0.5 | 1.02 | 9.03 | 1.44 | 1.03 | 1.21 | 1.04 |
|     | GSW(3) LR 0.9 | 1.01 | 42.90 | 3.62 | 1.12 | 1.72 | 1.08 |
|     | GSW(3) LR 0.99 | **1.00** | 122.27 | 5.86 | 2.51 | 2.65 | 1.44 |
|     | GSW(2) LR 0.01 | 1.02 | 6.20 | 1.41 | 1.09 | 1.20 | 1.09 |
|     | GSW(2) LR 0.1 | 1.02 | 6.55 | 1.42 | 1.08 | 1.21 | 1.09 |
|     | GSW(2) LR 0.5 | 1.01 | 10.50 | 1.55 | 1.08 | 1.28 | 1.09 |
|     | GSW(2) LR 0.9 | 1.01 | 40.43 | 3.36 | 1.12 | 1.62 | 1.09 |
|     | GSW(2) LR 0.99 | **1.00** | 118.90 | 5.72 | 2.45 | 2.58 | 1.41 |
|     | GSW(1) LR 0.01 | 1.01 | **1.00** | 6.36 | 1.08 | 2.87 | 1.09 |
|     | GSW(1) LR 0.1 | 1.01 | 1.14 | 6.34 | 1.08 | 2.85 | 1.08 |
|     | GSW(1) LR 0.5 | 1.01 | 3.19 | 6.35 | 1.08 | 2.86 | 1.08 |
|     | GSW(1) LR 0.9 | 1.01 | 21.76 | 6.33 | 1.08 | 2.85 | 1.08 |
|     | GSW(1) LR 0.99 | **1.00** | 99.57 | 6.33 | 2.08 | 2.85 | 1.25 |

## B.5. Sensitivity to the Low-Rank Parameter

The main experiments use a rank-$2d$ approximation for $n > 100$ as a computationally efficient default. To assess sensitivity to this choice, we repeat the $n = 200, d = 10$ quadratic-outcome experiment while varying the rank parameter $k$. This setting is representative of studying rank sensitivity because it uses the low-rank implementation and directly tests nonlinear covariate balance. Tables 13 and 14 show that the performance is stable over the tested range of $k$, supporting the use of moderate-rank approximations in the main experiments.

*Table 7.* Robustness and covariate moment imbalance ratios to the best design for $d = 30$ and $n = 200$, full table. For each column metric, numbers are divided by the minimum number. CR = complete randomization; Block = blocking; Match = pairwise matching; RR = rerandomization; LR = low rank. For GSW variant, the last parameter is the value of $\phi$.

| $n$ | Design | Robustness ($\downarrow$) | Covariate Moment ($\downarrow$) 1 | 2 | 3 | 4 | 5 |
|---|---|---|---|---|---|---|---|
| 200 | Bernoulli | **1.00** | 28.48 | 3.03 | 1.25 | 1.03 | **1.00** |
| | CR | **1.00** | 28.48 | 1.24 | 1.25 | 1.01 | 1.01 |
| | Block | 1.01 | 28.46 | 1.24 | 1.25 | **1.00** | 1.01 |
| | Match | 1.65 | 21.63 | 1.53 | 1.60 | 1.55 | 1.61 |
| | RR | 1.05 | 13.30 | 1.28 | 1.01 | 1.05 | 1.04 |
| | GSW($\infty$) LR 0.01 | 1.15 | 2.95 | **1.00** | 1.08 | 1.06 | 1.13 |
| | GSW($\infty$) LR 0.1 | 1.13 | 2.95 | 1.03 | 1.06 | 1.05 | 1.11 |
| | GSW($\infty$) LR 0.5 | 1.06 | 7.88 | 1.18 | 1.01 | 1.04 | 1.05 |
| | GSW($\infty$) LR 0.9 | 1.01 | 20.97 | 1.23 | 1.07 | 1.01 | **1.00** |
| | GSW($\infty$) LR 0.99 | **1.00** | 27.44 | 1.82 | 1.22 | **1.00** | **1.00** |
| | GSW(3) LR 0.01 | 1.07 | 7.13 | 1.95 | 1.01 | 1.04 | 1.05 |
| | GSW(3) LR 0.1 | 1.07 | 6.69 | 1.90 | 1.01 | 1.04 | 1.05 |
| | GSW(3) LR 0.5 | 1.05 | 9.87 | 2.25 | **1.00** | 1.03 | 1.03 |
| | GSW(3) LR 0.9 | 1.01 | 20.92 | 2.85 | 1.07 | 1.02 | **1.00** |
| | GSW(3) LR 0.99 | **1.00** | 27.22 | 3.02 | 1.21 | 1.03 | **1.00** |
| | GSW(2) LR 0.01 | 1.09 | 4.78 | 1.57 | 1.03 | 1.06 | 1.08 |
| | GSW(2) LR 0.1 | 1.08 | 4.95 | 1.60 | 1.03 | 1.05 | 1.07 |
| | GSW(2) LR 0.5 | 1.05 | 7.91 | 2.04 | 1.01 | 1.04 | 1.04 |
| | GSW(2) LR 0.9 | 1.01 | 18.89 | 2.76 | 1.04 | 1.02 | **1.00** |
| | GSW(2) LR 0.99 | **1.00** | 27.17 | 3.02 | 1.21 | 1.03 | **1.00** |
| | GSW(1) LR 0.01 | 1.12 | **1.00** | 3.10 | 1.05 | 1.13 | 1.10 |
| | GSW(1) LR 0.1 | 1.11 | 1.20 | 3.10 | 1.05 | 1.13 | 1.10 |
| | GSW(1) LR 0.5 | 1.10 | 3.19 | 3.09 | 1.04 | 1.12 | 1.08 |
| | GSW(1) LR 0.9 | 1.03 | 13.61 | 3.05 | 1.01 | 1.06 | 1.02 |
| | GSW(1) LR 0.99 | **1.00** | 25.61 | 3.05 | 1.17 | 1.03 | **1.00** |

*Table 8.* MSE ratios to the best design for $d = 10$ and $n = 200$ and $\sigma = 0.1$, full table. For each column metric, numbers are divided by the minimum number. CR = complete randomization; Block = blocking; Match = pairwise matching; RR = rerandomization; LR = low rank. For GSW variant, the last parameter is the value of $\phi$.

| Design | Linear | Quadratic | Cubic | Sinusoidal |
|---|---|---|---|---|
| Bernoulli | 52.35 | 3.17 | 3.14 | 10.38 |
| CR | 52.04 | 1.89 | 3.13 | 6.07 |
| Block | 46.25 | 1.82 | 2.93 | 5.42 |
| Match | 14.72 | **1.00** | 1.32 | 1.93 |
| RR | 10.95 | 1.90 | 1.42 | 1.85 |
| GSW($\infty$) LR 0.01 | 3.74 | 1.56 | 1.12 | **1.00** |
| GSW($\infty$) LR 0.1 | 4.29 | 1.60 | 1.14 | 1.08 |
| GSW($\infty$) LR 0.5 | 10.49 | 1.77 | 1.39 | 1.79 |
| GSW($\infty$) LR 0.9 | 33.08 | 2.01 | 2.35 | 4.58 |
| GSW($\infty$) LR 0.99 | 49.20 | 2.64 | 3.02 | 8.27 |
| GSW(3) LR 0.01 | 2.64 | 1.58 | 1.02 | 1.48 |
| GSW(3) LR 0.1 | 2.82 | 1.60 | 1.04 | 1.55 |
| GSW(3) LR 0.5 | 6.70 | 2.03 | 1.21 | 2.67 |
| GSW(3) LR 0.9 | 26.52 | 2.78 | 2.08 | 6.64 |
| GSW(3) LR 0.99 | 47.90 | 3.12 | 2.98 | 9.74 |
| GSW(2) LR 0.01 | 3.08 | 1.74 | 1.10 | 1.65 |
| GSW(2) LR 0.1 | 3.34 | 1.77 | 1.09 | 1.74 |
| GSW(2) LR 0.5 | 6.82 | 2.07 | 1.25 | 2.66 |
| GSW(2) LR 0.9 | 25.07 | 2.72 | 2.01 | 6.34 |
| GSW(2) LR 0.99 | 46.68 | 3.10 | 2.92 | 9.73 |
| GSW(1) LR 0.01 | **1.00** | 3.18 | **1.00** | 5.19 |
| GSW(1) LR 0.1 | 1.12 | 3.17 | **1.00** | 5.18 |
| GSW(1) LR 0.5 | 3.00 | 3.16 | 1.09 | 5.35 |
| GSW(1) LR 0.9 | 16.77 | 3.14 | 1.66 | 6.73 |
| GSW(1) LR 0.99 | 42.81 | 3.17 | 2.74 | 9.54 |

*Table 9.* MSE ratios to the best design for $d = 10$ and $n = 500$ and $\sigma = 0.1$, full table. For each column metric, numbers are divided by the minimum number. CR = complete randomization; Block = blocking; Match = pairwise matching; RR = rerandomization; LR = low rank. For GSW variant, the last parameter is the value of $\phi$.

| Design | Linear | Quadratic | Cubic | Sinusoidal |
|---|---|---|---|---|
| Bernoulli | 113.16 | 3.97 | 3.28 | 11.88 |
| CR | 113.75 | 2.34 | 3.30 | 6.94 |
| Block | 87.20 | 2.05 | 2.73 | 5.41 |
| Match | 24.78 | **1.00** | 1.11 | 1.74 |
| RR | 23.84 | 2.35 | 1.47 | 2.08 |
| GSW($\infty$) LR 0.01 | 5.55 | 1.95 | 1.11 | **1.00** |
| GSW($\infty$) LR 0.1 | 6.13 | 1.98 | 1.11 | 1.03 |
| GSW($\infty$) LR 0.5 | 12.60 | 2.13 | 1.25 | 1.44 |
| GSW($\infty$) LR 0.9 | 48.41 | 2.37 | 1.98 | 3.62 |
| GSW($\infty$) LR 0.99 | 100.37 | 2.91 | 3.02 | 7.94 |
| GSW(3) LR 0.01 | 3.22 | 1.86 | 1.02 | 1.31 |
| GSW(3) LR 0.1 | 3.50 | 1.90 | 1.03 | 1.36 |
| GSW(3) LR 0.5 | 7.44 | 2.23 | 1.13 | 2.02 |
| GSW(3) LR 0.9 | 34.99 | 3.13 | 1.70 | 5.42 |
| GSW(3) LR 0.99 | 92.94 | 3.83 | 2.88 | 10.31 |
| GSW(2) LR 0.01 | 4.96 | 2.16 | 1.09 | 1.66 |
| GSW(2) LR 0.1 | 5.24 | 2.18 | 1.10 | 1.72 |
| GSW(2) LR 0.5 | 8.54 | 2.37 | 1.17 | 2.20 |
| GSW(2) LR 0.9 | 32.64 | 3.08 | 1.67 | 5.08 |
| GSW(2) LR 0.99 | 90.59 | 3.77 | 2.83 | 10.11 |
| GSW(1) LR 0.01 | **1.00** | 3.97 | **1.00** | 5.75 |
| GSW(1) LR 0.1 | 1.11 | 3.97 | 1.01 | 5.74 |
| GSW(1) LR 0.5 | 2.75 | 3.96 | 1.04 | 5.89 |
| GSW(1) LR 0.9 | 17.77 | 3.97 | 1.35 | 6.62 |
| GSW(1) LR 0.99 | 76.08 | 3.92 | 2.53 | 9.80 |

*Table 10.* MSE ratios to the best design for $d = 10$ and n = 200, $\sigma = 0.5$, norm, full table. For each column metric, numbers are divided by the minimum number. CR = complete randomization; Block = blocking; Match = pairwise matching; RR = rerandomization; LR = low rank. For GSW variant, the last parameter is the value of $\phi$.

| Design | Linear | Quadratic | Cubic | Sinusoidal |
|---|---|---|---|---|
| Bernoulli | 16.68 | 3.14 | 3.14 | 3.79 |
| CR | 16.53 | 1.88 | 3.13 | 2.49 |
| Block | 14.74 | 1.81 | 2.93 | 2.31 |
| Match | 5.23 | **1.00** | 1.32 | 1.28 |
| RR | 4.03 | 1.89 | 1.41 | 1.25 |
| GSW($\infty$) LR 0.01 | 1.84 | 1.56 | 1.12 | **1.00** |
| GSW($\infty$) LR 0.1 | 2.00 | 1.59 | 1.14 | 1.02 |
| GSW($\infty$) LR 0.5 | 3.92 | 1.76 | 1.39 | 1.23 |
| GSW($\infty$) LR 0.9 | 10.80 | 2.00 | 2.35 | 2.07 |
| GSW($\infty$) LR 0.99 | 15.65 | 2.61 | 3.02 | 3.16 |
| GSW(3) LR 0.01 | 1.51 | 1.57 | 1.02 | 1.14 |
| GSW(3) LR 0.1 | 1.55 | 1.59 | 1.04 | 1.16 |
| GSW(3) LR 0.5 | 2.74 | 2.01 | 1.21 | 1.50 |
| GSW(3) LR 0.9 | 8.78 | 2.76 | 2.08 | 2.65 |
| GSW(3) LR 0.99 | 15.28 | 3.10 | 2.98 | 3.58 |
| GSW(2) LR 0.01 | 1.65 | 1.74 | 1.10 | 1.19 |
| GSW(2) LR 0.1 | 1.71 | 1.77 | 1.09 | 1.22 |
| GSW(2) LR 0.5 | 2.77 | 2.06 | 1.25 | 1.48 |
| GSW(2) LR 0.9 | 8.36 | 2.69 | 2.01 | 2.59 |
| GSW(2) LR 0.99 | 14.94 | 3.07 | 2.92 | 3.57 |
| GSW(1) LR 0.01 | **1.00** | 3.15 | **1.00** | 2.23 |
| GSW(1) LR 0.1 | 1.04 | 3.14 | **1.00** | 2.24 |
| GSW(1) LR 0.5 | 1.62 | 3.13 | 1.09 | 2.30 |
| GSW(1) LR 0.9 | 5.78 | 3.11 | 1.66 | 2.67 |
| GSW(1) LR 0.99 | 13.73 | 3.14 | 2.74 | 3.54 |

*Table 11.* MSE ratios to the best design for $d = 30$ and n = 200, $\sigma = 0.1$, full table. For each column metric, numbers are divided by the minimum number. CR = complete randomization; Block = blocking; Match = pairwise matching; RR = rerandomization; LR = low rank. For GSW variant, the last parameter is the value of $\phi$.

| Design | Linear | Quadratic | Cubic | Sinusoidal |
|---|---|---|---|---|
| Bernoulli | 19.99 | 1.84 | 2.53 | 15.72 |
| CR | 19.82 | 1.19 | 2.51 | 4.27 |
| Block | 19.78 | 1.18 | 2.52 | 4.29 |
| Match | 11.42 | **1.00** | 1.83 | 2.52 |
| RR | 9.39 | 1.20 | 1.68 | 2.19 |
| GSW($\infty$) LR 0.01 | 3.03 | 1.09 | 1.16 | **1.00** |
| GSW($\infty$) LR 0.1 | 3.03 | 1.11 | 1.17 | 1.01 |
| GSW($\infty$) LR 0.5 | 7.67 | 1.17 | 1.54 | 2.04 |
| GSW($\infty$) LR 0.9 | 16.33 | 1.25 | 2.24 | 4.84 |
| GSW($\infty$) LR 0.99 | 19.47 | 1.56 | 2.48 | 10.81 |
| GSW(3) LR 0.01 | 6.99 | 1.56 | 1.48 | 8.78 |
| GSW(3) LR 0.1 | 6.69 | 1.55 | 1.45 | 8.62 |
| GSW(3) LR 0.5 | 9.60 | 1.64 | 1.69 | 10.53 |
| GSW(3) LR 0.9 | 16.59 | 1.79 | 2.26 | 14.34 |
| GSW(3) LR 0.99 | 19.76 | 1.83 | 2.51 | 15.54 |
| GSW(2) LR 0.01 | 4.68 | 1.48 | 1.29 | 6.79 |
| GSW(2) LR 0.1 | 4.90 | 1.49 | 1.32 | 6.94 |
| GSW(2) LR 0.5 | 7.87 | 1.60 | 1.55 | 9.34 |
| GSW(2) LR 0.9 | 15.55 | 1.77 | 2.17 | 13.71 |
| GSW(2) LR 0.99 | 19.47 | 1.83 | 2.48 | 15.49 |
| GSW(1) LR 0.01 | **1.00** | 1.86 | **1.00** | 11.97 |
| GSW(1) LR 0.1 | 1.21 | 1.86 | 1.01 | 12.01 |
| GSW(1) LR 0.5 | 3.24 | 1.84 | 1.18 | 12.37 |
| GSW(1) LR 0.9 | 11.79 | 1.85 | 1.88 | 14.09 |
| GSW(1) LR 0.99 | 18.72 | 1.85 | 2.41 | 15.59 |

*Table 12.* MSE ratios to the best design for $d = 10$, $n = 100$, and $\sigma = 0.1$, with additional kernel-rerandomization baselines. For each column metric, numbers are divided by the minimum number. CR = complete randomization; Block = blocking; Match = pairwise matching; RR = rerandomization. For each GSW variant, the last parameter is the value of $\phi$.

| Design | Linear ($\downarrow$) | Quadratic ($\downarrow$) | Cubic ($\downarrow$) | Sinusoidal ($\downarrow$) |
|---|---|---|---|---|
| Bernoulli | 27.42 | 5.21 | 3.32 | 12.18 |
| CR | 27.49 | 3.14 | 3.31 | 7.09 |
| Block | 26.05 | 3.05 | 3.23 | 6.83 |
| Match | 9.16 | 1.87 | 1.55 | 2.75 |
| RR | 5.96 | 3.13 | 1.51 | 2.22 |
| RR (quadratic kernel) | 15.68 | 1.97 | 2.33 | 4.23 |
| RR ($L_2$ discrepancy kernel) | 9.19 | 2.55 | 1.79 | 2.64 |
| GSW($\infty$) 0.01 | 2.25 | 1.20 | 1.08 | **1.00** |
| GSW($\infty$) 0.1 | 2.82 | 1.61 | 1.19 | 1.22 |
| GSW($\infty$) 0.5 | 7.24 | 2.61 | 1.61 | 2.51 |
| GSW($\infty$) 0.9 | 19.74 | 3.43 | 2.69 | 6.29 |
| GSW($\infty$) 0.99 | 26.46 | 4.59 | 3.27 | 10.49 |
| GSW(3) 0.01 | 2.36 | 1.79 | **1.00** | 1.80 |
| GSW(3) 0.1 | 2.78 | 2.09 | 1.09 | 2.09 |
| GSW(3) 0.5 | 5.98 | 3.26 | 1.47 | 3.92 |
| GSW(3) 0.9 | 17.57 | 4.64 | 2.51 | 8.79 |
| GSW(3) 0.99 | 26.01 | 5.14 | 3.23 | 11.71 |
| GSW(2) 0.01 | 1.78 | **1.00** | 1.18 | 1.25 |
| GSW(2) 0.1 | 2.09 | 1.41 | 1.19 | 1.49 |
| GSW(2) 0.5 | 4.90 | 2.87 | 1.43 | 3.19 |
| GSW(2) 0.9 | 16.41 | 4.50 | 2.42 | 8.26 |
| GSW(2) 0.99 | 25.75 | 5.12 | 3.21 | 11.61 |
| GSW(1) 0.01 | **1.00** | 5.27 | 1.09 | 6.16 |
| GSW(1) 0.1 | 1.13 | 5.26 | 1.10 | 6.20 |
| GSW(1) 0.5 | 3.05 | 5.27 | 1.28 | 6.56 |
| GSW(1) 0.9 | 13.48 | 5.20 | 2.17 | 8.93 |
| GSW(1) 0.99 | 25.01 | 5.20 | 3.15 | 11.60 |

*Table 13.* Sensitivity of GSW(2) with $\alpha_1 = 2/3, \alpha_2 = 1/3$ to the low-rank parameter $k$ for $n = 200$, $d = 10$, $\phi = 0.5$, and quadratic outcomes with $\sigma = 0.1$. Results are averaged over 10,000 GSW simulations. The column $20d$ corresponds to the full-rank implementation.

| Rank level $k$ | $d$ | $2d$ | $4d$ | $8d$ | $20d$ (full) |
|---|---|---|---|---|---|
| Covariate 1st Moment ($\downarrow$) | 9.63 | 4.12 | 2.97 | 2.86 | 2.83 |
| Covariate 2nd Moment ($\downarrow$) | 15.57 | 10.23 | 7.28 | 5.66 | 5.65 |
| MSE ($\downarrow$) | 0.31 | 0.24 | 0.18 | 0.14 | 0.14 |
| Runtime (s) | 18.00 | 30.05 | 64.38 | 392.66 | 2578.38 |

*Table 14.* Sensitivity of GSW($\infty$) to the low-rank parameter $k$ for $n = 200$, $d = 10$, $\phi = 0.5$, and quadratic outcomes with $\sigma = 0.1$. Results are averaged over 10,000 GSW simulations. The column $20d$ corresponds to the full-rank implementation.

| Rank level $k$ | $d$ | $2d$ | $4d$ | $8d$ | $20d$ (full) |
|---|---|---|---|---|---|
| Covariate 1st Moment ($\downarrow$) | 12.85 | 6.41 | 4.69 | 4.29 | 4.46 |
| Covariate 2nd Moment ($\downarrow$) | 11.51 | 10.45 | 8.74 | 7.40 | 7.01 |
| MSE ($\downarrow$) | 0.22 | 0.21 | 0.19 | 0.16 | 0.15 |
| Runtime (s) | 15.65 | 26.11 | 54.57 | 381.02 | 2558.77 |

