# OpenReview forum: "Nonlinear Covariate Balance in Experimental Design"
_ICML.cc/2026/Conference — ICML 2026 regular_

### Official Review · Reviewer_VUEu · 2026-03-13

**Soundness:** 3
**Presentation:** 3
**Significance:** 3
**Originality:** 3
**Overall Recommendation:** 5
**Confidence:** 2

**Summary:**

The paper proposes a nonlinear extension of the Gram-Schmidt Walk (GSW) design for randomized controlled trials. It leverages a Gram matrix representation and a feature map to balance nonlinear covariate structures (e.g., higher-order moments and smooth functions) without explicitly constructing high-dimensional features. To address computational bottlenecks associated with large Gram matrices, the authors introduce a low-rank approximation approach, reducing the runtime from $\mathcal{O}(n^3)$ to $\mathcal{O}(n^2k)$. The proposed method is empirically evaluated against standard randomization schemes across various linear and nonlinear outcome models.

**Compliance With Llm Reviewing Policy:**

Affirmed.

**Final Justification:**

The authors provided a clear, transparent, and satisfying response that fully addressed my initial concerns. Their clarifications and the additional empirical results reinforced my positive assessment of the paper's soundness and overall significance. I maintain my original score and recommend accepting this paper.

**Key Questions For Authors:**

1. In a real-world experimental design setting, the true outcome-generating model is unknown beforehand. Are there practical heuristics or data-driven strategies to safely select the balance-robustness parameter $\phi$ and the feature map weights $\alpha_k$ (e.g., deciding between GSW(2), GSW(3), or GSW($\infty$)) prior to the experiment?

2. While the integration of the Nyström method for low-rank approximation successfully reduces the algorithmic runtime to $O(n^2k)$, quadratic complexity with respect to the sample size $n$ might still be computationally prohibitive for massive A/B testing platforms. Is this design primarily intended for moderate-sized trials, or are there further optimizations for larger scales?

3. The empirical evaluation compares the proposed method against classical designs (Bernoulli, Complete Randomization, Blocking, Matching, Re-randomization). The authors excluded the optimal kernel allocation design by Kallus (2018) due to its prohibitive runtime. Are there any other computationally tractable kernel-based or nonlinear experimental design baselines that could serve as a fairer comparison to evaluate the specific performance gains of the GSW approach?

**Limitations:**

The paper could be improved by explicitly discussing the practical limitations of the algorithm itself, specifically, the challenge of a priori hyperparameter selection (choosing $\phi$ and the truncation level of the polynomial design) without having access to the outcomes in a blind randomized controlled trial. Acknowledging this limitation would provide better guidance for practitioners.

**Strengths And Weaknesses:**

# Strengths

**- Soundness:** The submission is supported by a comprehensive empirical demonstration. The simulations thoroughly cover different outcome generating processes (linear, quadratic, cubic, sinusoidal) and clearly demonstrate the benefits of the proposed nonlinear variants (e.g., GSW(2), GSW(3), GSW($\infty$)) over classical baselines. Furthermore, the use of the Nyström method for low-rank approximation effectively bridges the gap between theoretical properties and computational feasibility.

**- Presentation:** The paper is clearly written. The authors clearly identify a significant limitation in the standard linear GSW, illustrating that balancing only linear covariates can lead to poor precision when the outcome model involves nonlinear relationships.

**- Significance:** The work addresses a well-motivated problem. By effectively balancing nonlinear covariate structures, it offers a highly practical solution to improve precision in real-world experimental designs.

**- Originality:** The authors propose an elegant methodology. Integrating the kernel trick into the GSW framework via a Gram matrix is a neat and practical solution to handle infinite-dimensional feature expansions implicitly.

# Weaknesses:
## 1. A priori Hyperparameter Selection:
The performance of Nonlinear GSW heavily depends on the choice of the balance-robustness parameter $\phi$ and the feature map weights $\alpha_k$. In a real-world experimental design setting, the practitioner does not know the true outcome-generating model beforehand.

## 2. Scalability to Massive RCTs:
While the low-rank approximation effectively reduces the runtime to $\mathcal{O}(n^2k)$, a quadratic complexity with respect to the sample size $n$ can still be computationally prohibitive for large-scale digital experiments (where $n$ can easily reach millions).

---

> ### Author Rebuttal · Authors · 2026-03-30
>
> We thank the reviewer for the careful reading and constructive suggestions.
>
> **A priori hyperparameter selection**
>
> We agree that the true outcome-generating model is unknown in practice. The choice of hyperparameters mainly relies on prior knowledge about the outcome model.  Thus, our goal is to improve estimation precision when the selected feature class (e.g., low-degree polynomials of covariates) is informative, while remaining robust when it is not informative or misspecified.
>
> The feature weight $\alpha_k$ determines how much priority is placed on balancing the $k$th covariate moment.
> We recommend placing a large weight on the first moment (i.e., choosing $\alpha_1$ relatively large), since balance on raw covariates is important in a wide range of settings. If low-order nonlinearities or interactions are expected, we recommend a low-degree polynomial design such as GSW(2) or GSW(3). We find that even a small positive $\alpha_2$ or $\alpha_3$ can greatly reduce second- or third-moment imbalance than the orignal GSW, with only a minor increase in first-moment imbalance. We use GSW($\infty$) as a flexible choice when one wants a broader class of smooth functions, which includes all moments and is considerably more expressive.
>
> For balance-robustness tradeoff parameter $\phi$, we recommend a conservative default such as $\phi \ge 0.5$ unless the practitioner is highly confident that the selected feature class is very informative to the outcome, following the guidance of Harshaw et al. (2024) for the original GSW. This keeps the robustness guarantee within a factor of two of Bernoulli randomization even under model misspecification (for example, if the covariates and outcomes are not correlated).
>
> **Scalability to massive RCTs**
>
> Our current method is mainly intended for moderate-sized trials, roughly hundreds to low thousands of units, rather than million-scale digital experiments. Our targeted regime is the same as the original GSW, whose runtime is also quadratic in the sample size.
>
> Further reducing the runtime for large-scale experiments is an interesting direction for future work. One promising approach is to combine our kernel/Gram-matrix viewpoint with the Balancing Walk Design of Arbour et al. (2022), which can be seen as an online, linear-time analogue of GSW, albeit with a slightly weaker covariate-balance guarantee.
>
> **Computationally tractable kernel-based design baselines**
>
> Thank you for this helpful suggestion. Beyond the classical baselines in the paper, the most relevant tractable nonlinear comparison we found is the kernel-discrepancy rerandomization method of Li et al. (2019). We ran an additional experiment under the same setup as our Section 7 ($n=100$, $d=10$, and the same four outcome classes).  We compare GSW(1), GSW(2), GSW(3), and GSW($\infty$) against three rerandomization (RR) baselines: classical RR with Mahalanobis distance, kernel-discrepancy RR with the quadratic kernel, and RR with the $L_2$-discrepancy kernel. For GSW, we set $\phi=0.01$ to mainly emphasize covariate balance. We report MSE, normalized within each outcome class by dividing by the smallest MSE in that row, in the following table.
>
> In all four outcome classes, the best-performing method is always a GSW variant, and both kernel-discrepancy RR baselines are consistently worse than the best GSW choice. One explanation might be that GSW constructs assignments that cancel imbalance in the relevant nonlinear feature space, whereas RR only filters assignments according to one scalar discrepancy. We will add this comparison.
>
> |            |   GSW(1) |   GSW(2) |   GSW(3) |   GSW($\infty$) |   RR |   RR (quad) |   RR ($L_2$) |
> |:-----------|:---------:|:---------:|:---------:|:----------------:|:-----:|:-----------------:|:------------------:|
> | linear     |     1.00 |     1.78 |     2.36 |            2.25 | 5.96 |            15.68 |              9.19 |
> | quadratic  |     5.27 |     1.00 |     1.79 |            1.20 | 3.13 |             1.97 |              2.55 |
> | cubic      |     1.09 |     1.18 |     1.00 |            1.08 | 1.51 |             2.33 |              1.79 |
> | sinusoidal |     6.16 |     1.25 |     1.80 |            1.00 | 2.22 |             4.23 |              2.64 |
>
> **References**
>
> Arbour, D., Dimmery, D., Mai, T., and Rao, A. Online balanced experimental design. In International Conference on Machine Learning, pp. 844–864. PMLR, 2022.
>
> Li, Yiou, Xiao Huang, and Lulu Kang. "A Discrepancy-Based Design for A/B Testing Experiments." arXiv preprint arXiv:1901.08984 (2019).

---

> > ### Author Rebuttal · Reviewer_VUEu · 2026-04-03
> >
> > Thank the authors for their detailed and transparent rebuttal. My concerns have been fully resolved. I will maintain my original score.

---

> > > ### Author Response · Authors · 2026-04-04
> > >
> > > Thank you very much for your thoughtful review and follow-up. We truly appreciate your positive and detailed comments. We are glad that our rebuttal addressed your concerns.

---

### Official Review · Reviewer_Hgr5 · 2026-03-13

**Soundness:** 3
**Presentation:** 2
**Significance:** 3
**Originality:** 2
**Overall Recommendation:** 3
**Confidence:** 3

**Summary:**

This paper studies randomized experimental design for average treatment effect estimation in two-arm RCTs. Building on the Gram-Schmidt Walk (GSW), the paper proposes a nonlinear extension that balances richer nonlinear functions of covariates through a feature-map. Overall, the problem is meaningful and the proposed extension is technically coherent, though the novelty is better viewed as a natural and well-executed extension of GSW rather than a fundamentally new design paradigm.

**Compliance With Llm Reviewing Policy:**

Affirmed.

**Final Justification:**

Thanks for the rebuttal and all efforts of ACs, SACs, PCs, ... After reading the paper and the response more carefully, I will keep my score at 3. I think the paper is technically coherent, and I appreciate the effort to extend GSW beyond linear covariate balance. The rebuttal also helps clarify the intended intuition and contribution.

However, my overall assessment of originality has not changed much. To me, the main step from linear GSW to a nonlinear or kernelized version feels natural, especially given the existing literature on kernel-based balancing. For this reason, I currently view the paper as a meaningful but fairly incremental extension, rather than a fundamentally new design idea.

I also still think the presentation makes the work harder to evaluate than necessary. The intuition for GSW and the main methodological point should appear earlier and more plainly. Because there are only two reviews and the other review is substantially more positive, I would encourage the AC to obtain one additional opinion if possible, or to take a closer personal look before making a final decision.

**Key Questions For Authors:**

Q1: Could the abstract and introduction explain GSW more explicitly for readers outside the experimental-design literature?
Q2: Can the authors state the main methodological idea earlier and more plainly, before introducing the more technical feature-space formalism?
Q3: How sensitive is performance to the choice of feature weights​, the choice of nonlinear function class, and the low-rank approximation level?

**Limitations:**

The paper studies a useful design problem, but the current version underplays the intuition early on and may overassume the reader's familiarity with GSW.

**Strengths And Weaknesses:**

Strengths:
S1: Sound motivation. The paper correctly identifies that balancing only linear functions of covariates may be insufficient when treatment outcomes depend on nonlinear covariate structure
S2: The methodological idea is clean: instead of explicitly constructing high-dimensional nonlinear covariate expansions, the paper works directly with the Gram matrix induced by a nonlinear feature map and runs Gram GSW there.

Weaknesses:
W1: The paper relies heavily on GSW from the abstract onward without quickly explaining it for non-specialist readers. Since GSW is central to the manuscript, a one-sentence intuitive explanation should appear earlier.
W2: The introduction motivates nonlinear balance well, but the precise methodological contribution is stated somewhat late.
W3: The main contribution is substantial but still fairly incremental relative to prior GSW.

---

> ### Author Rebuttal · Authors · 2026-03-31
>
> We thank the reviewer for the careful reading and constructive suggestions. We appreciate the positive comments on the sound motivation and the clean methodological idea.
>
> **GSW explanation**
>
> We thank the reviewer for suggesting adding an intuitive GSW explanation earlier.
>
> Intuitively, GSW is a restricted-randomization procedure that constructs the assignment vector via a carefully guided random walk, freezing one unit at a time while keeping the induced covariate imbalance small. We will add this in the Introduction when we first introduce GSW.
>
> **Our methodology**
>
> We will state the main methodological idea earlier and more plainly in the Introduction.
>
> Our main methodological idea is to reinterpret GSW through the unit-by-unit Gram matrix of the covariates, which is always of dimensions $n \times n$. A naive extension of GSW to nonlinear covariate functions requires constructing high- or even infinite-dimensional nonlinear features. Using the $n$-by-$n$ Gram matrix avoids such an explicit construction. This reinterpretation also enables us to prove finite-sample guarantees for causal estimation and propose a low-rank GSW for faster computation.
>
> **Contribution comparison with GSW**
>
> We agree that our work builds on GSW. However, extending its linear-feature guarantees to nonlinear function classes is nontrivial, and our main contribution is to make this extension possible with new finite-sample theory and a low-rank algorithm. In many applications, covariate-outcome relationships are nonlinear, and balancing nonlinear functions of covariates improves causal estimation.
>
> Specifically, the challenges include: explicit higher-order features can greatly increase dimensionality, balancing many nonlinear features may worsen balance on linear features, and it is unclear whether the statistical guarantees of GSW continue to hold and which nonlinear features should be balanced.
> Our contribution is to resolve these issues in a principled way. We establish new finite-sample covariance, discrepancy, and MSE guarantees for nonlinear feature maps, and quantify when including higher-order terms improves estimation accuracy. We also develop a scalable low-rank implementation and analyze how approximation error propagates.
>
> **Sensitivity to hyperparameter choices**
>
> Performance does depend on these choices, but the dependence is structured. In both the theory and our experiments, moderate changes in the feature weights, function class, and rank lead to predictable tradeoffs, and reasonable default choices already perform well.
>
> **Feature weights.**
> The feature weights affect covariate balance smoothly (Lemma 4.2). A larger feature weight $\alpha_j$ enforces a better balance on the $j$th moment of the covariate imbalance. We also support the theory with a numerical simulation with $n=100$ and $d=10$. We vary the weights $\alpha_1,\alpha_2$ and the balance-robustness tradeoff parameter $\phi$, and measure the resulting first- and second-moment imbalance (see Figure 1 in https://anonymous.4open.science/r/icml2026-rebuttal-materials/README.md).
>
> In practice, we recommend choosing a relatively large $\alpha_1$, since balancing linear covariate terms is important in many applications. We also recommend using small positive values for $\alpha_2$ and $\alpha_3$: even modest weights can substantially improve balance on the second and third moments, while the loss in first-moment balance is typically gradual rather than sharp.
>
> **Function class.**
> The dependence of performance on the choice of nonlinear function class is characterized by how well the outcome can be approximated within the selected class, with both small approximation error and controlled complexity (Proposition 5.1).
> Low-degree polynomials are a conservative choice when mild nonlinearities are expected. The exponential kernel is more expressive and a flexible choice when a richer, smooth, nonlinear structure is plausible.
>
> **Low-rank level.**
> The rank parameter k affects performance only through the kernel approximation error $\epsilon_k = \|K-\widetilde K\|_2$, which typically decreases as k increases. Thus, smaller k gives lower computational cost but a looser guarantee; as k increases, $\epsilon_k$ decreases, and the low-rank GSW smoothly approaches the exact full-kernel GSW.
>
> Performance is usually stable once the leading eigenspace of $K$ is captured. In our experiments, $k=4d$ or $8d$ is already enough. We support this in a numerical simulation for $n=200,d=10$ where we vary $k$ from $d$ to $20d$ (Tables 1 and 2 in https://anonymous.4open.science/r/icml2026-rebuttal-materials/README.md).

---

> > ### Author Rebuttal · Reviewer_Hgr5 · 2026-04-04
> >
> > Thank you for your response

---

> > > ### Author Response · Authors · 2026-04-04
> > >
> > > Thank you again for your thoughtful feedback. We are glad that our rebuttal addressed your concerns. If appropriate, we would be grateful if you might consider adjusting your score to reflect your updated view of the paper. We very much appreciate your time and consideration.

---

### Decision · Program_Chairs · 2026-04-30

**Decision:**

Accept (regular)

**Comment:**

Authors tackle the problem of splitting units with covariate features into treatment and control for experimental design. It is classically known that the Bernoulli 1/2 assignment is robust but it creates covariate imbalance due to randomness at lower samples. In a seminal work Harshaw et. al. 2024 showed that total error with respect to the true treatment effect estimate is given by two terms: 1) Covariate balance captured as a norm of multiplying the assignment vectors (with entries $+1,-1$) with the feature matrix and 2) robustness which is related to the spectral norm of the covariant matrix of the assignment vector.

Harshaw et al. 2024 proposed a Gram Schmidt Walk based that optimizes the tradeoff between both based on a random walk inspired from discrepancy minimization literature in theoretical computer science. However, that paper addressed outcomes that are linear functions of features. The extension to non linear features remained unclear. Current paper aims to fix this gap. Here are the reviewer concerns,

1) Paper executes very well on the generalization of GSW to non linear features focusing on executing the Gram schmidt walk based on only the Kernel and not on the features themselves and provides guarantees.

2) Other reviewer points out that this could be viewed as a natural extension to Harshaw's paper.

Paper's execution is great and the fact that it addresses a noted shortcoming of the prior work on non linear features and authors also compared it to another baseline (RR) based on kernel discrepancy during the rebuttal.

However, from a technical point of view, authors observed that original GSW algorithm can be executed with kernels and this does not seem to a conceptual leap. There is clear technical working out of the guarantees which is not very hard given the prior results about Kernels and GSW.

Accept if there is room in the program